# Distinct effects of complement and of NLRP3- and non-NLRP3 inflammasomes for choroidal neovascularization

**Jakob Malsy[1,2], Andrea C Alvarado[1], Joseph O Lamontagne[1], Karin Strittmatter[1], Alexander G Marneros[1]***

[1]Cutaneous Biology Research Center, Massachusetts General Hospital and Harvard Medical School, Boston, United States; [2]Department of Ophthalmology, University of Halle, Halle, Germany

**Abstract** NLRP3 inflammasome activation and complement-mediated inflammation have been implicated in promoting choroidal neovascularization (CNV) in age-related macular degeneration (AMD), but central questions regarding their contributions to AMD pathogenesis remain unanswered. Key open questions are (1) whether NLRP3 inflammasome activation mainly in retinal pigment epithelium (RPE) or rather in non-RPE cells promotes CNV, (2) whether inflammasome activation in CNV occurs via NLRP3 or also through NLRP3-independent mechanisms, and (3) whether complement activation induces inflammasome activation in CNV. Here we show in a neovascular AMD mouse model that NLRP3 inflammasome activation in non-RPE cells but not in RPE cells promotes CNV. We demonstrate that both NLRP3-dependent and NLRP3-independent inflammasome activation mechanisms induce CNV. Finally, we find that complement and inflammasomes promote CNV through independent mechanisms. Our findings uncover an unexpected role of non-NLRP3 inflammasomes for CNV and suggest that combination therapies targeting inflammasomes and complement may offer synergistic benefits to inhibit CNV.

**\*For correspondence:**
amarneros@mgh.harvard.edu

**Competing interests:** The authors declare that no competing interests exist.

## Introduction

Inflammasomes are multiprotein complexes that lead to the proteolytic activation of proinflammatory IL-1β and IL-18 through the catalytic activity of caspase-1. Canonical inflammasome activation is initiated by cytosolic pattern recognition receptors (PRRs) when exposed to specific triggers (either microbe-derived pathogen-associated molecular patterns [PAMPs] or host-derived danger-associated molecular patterns [DAMPs]). These PRRs include the family of NOD-like receptors (NLRP1, NLRP3, NLRP6, NLRC4), or AIM2, IFI16, and pyrin. Upon sensing specific stimuli, PRRs oligomerize and form together with the ASC adaptor protein scaffolds that activate caspase-1. This results in autoproteolytic activation of pro-caspase-1 that leads to the release of their effector domains caspase-1 p10 and p20, which are required for the proteolytic processing of pro-IL-1β and pro-IL-18 into their active mature forms that promote inflammation and angiogenesis (*Guo et al., 2015*; *Latz et al., 2013*; *Schroder and Tschopp, 2010*).

Of note, a non-canonical inflammasome has been described in which bacterial lipopolysaccharide (LPS) activates caspase-11 (caspase-4/–5 in humans) to induce pyroptotic cell death (*Kayagaki et al., 2011*).

Recent studies implicated canonical inflammasome activation in AMD pathogenesis. However, these studies have focused on the NLRP3 inflammasome and did not consider a potential contribution of other PRRs for inflammasome activation in AMD. The reason for this very narrow focus on NLRP3 as a PRR that initiates inflammasome activation in AMD is that various stimuli that are well-established risk factors for AMD, including increased oxidative stress or lipid accumulations, are

known activators of the NLRP3 inflammasome (*Guo et al., 2015*; *Latz et al., 2013*; *Schroder and Tschopp, 2010*). This has led to the assumption that AMD risk factors promote NLRP3 inflammasome activation, which further exacerbates AMD pathologies through activation of proinflammatory cytokines, such as IL-1β that is known to stimulate inflammatory angiogenesis (*Lavalette et al., 2011*).

Indeed, recent studies provided some evidence that NLRP3 inflammasome activation occurs in human AMD and promotes AMD-like pathologies in animal models as well (*Doyle et al., 2012*; *Marneros, 2013*; *Marneros, 2016*; *Tarallo et al., 2012*). A study by Tarallo et al reported that NLRP3 is expressed in RPE cells in human non-exudative AMD and that targeting the NLRP3 inflammasome or IL-18 could inhibit RPE-degeneration (*Tarallo et al., 2012*). This study used a model of acute RPE degeneration caused by *Alu* RNA accumulation or DICER loss as an experimental model for non-exudative AMD, which does not reflect many of the progressive age-dependent pathologies seen in human non-exudative AMD. A study by Doyle et al reported that NLRP3 inflammasome activation occurs in macrophages and results in IL-18 activation that inhibits laser-induced CNV (*Doyle et al., 2012*). This study used laser-induced CNV as a model for neovascular AMD, which is more of an acute injury model and not reflective of the progressive age-dependent development of CNV seen in human neovascular AMD that occurs without an acute injury (*He and Marneros, 2013*). Moreover, multiple other groups could not detect an effect of IL-18 on laser-induced CNV (*Hirano et al., 2014*). We also tested the role of IL-18 in a novel genetic mouse model of neovascular AMD, *Vegfa*^hyper mice, in which we observed that increased VEGF-A levels in the RPE resulted in increased inflammasome activation products in RPE/choroids of these mice as well as in spontaneous CNV formation in an age-dependent manner without experimental injury (*Ablonczy et al., 2014*; *Marneros, 2013*; *Marneros, 2016*). Thus, we regard *Vegfa*^hyper mice as a more disease-relevant animal model of neovascular AMD than the laser-injury CNV model. We showed that inactivation of *Il18* does not significantly affect CNV lesion formation in *Vegfa*^hyper mice, whereas inactivation of *Nlrp3*, *Casp1*, or *Il1r* potently inhibits CNV lesion formation in these mice (*Marneros, 2013*; *Marneros, 2016*). These findings provide evidence that activation of IL-1β through the NLRP3 inflammasome promotes VEGF-A-induced CNV in mice, whereas IL-18 has no major effect. Notably, inflammasome activation in RPE/choroids of *Vegfa*^hyper mice was associated with the accumulation of known triggers of the inflammasome in CNV lesions, including increased markers of oxidative stress and sub-RPE lipid deposits (*Marneros, 2013*). In this *Vegfa*^hyper mouse model of AMD, spontaneous CNV lesion formation can be accurately quantified already at a young age (here quantified at 6 weeks of age). These CNV lesions enlarge with progressive age and can merge with adjacent lesions to form larger confluent areas of CNV (*Ablonczy et al., 2014*; *Marneros, 2013*; *Marneros, 2016*), similarly as observed in human neovascular AMD where multiple adjacent ingrowth sites of neovessels from choriocapillaris through Bruch's membrane can merge to form larger CNV lesions (*Grossniklaus and Green, 2004*).

The findings from previous studies regarding the role of IL-18 for AMD have resulted in a controversy about whether inflammasomes play a role in AMD pathogenesis at all. For example, NLRP3 protein was not detected in the RPE of human eyes with AMD in a recent study that tested only a limited number of antibodies, questioning whether inflammasome activation occurs in the RPE at all (*Kosmidou et al., 2018*). However, this study did not investigate whether actual inflammasome activation occurs in the RPE (cleaved caspase-1 p10 or p20 levels or IL-1β secretion were not assessed, which serve as inflammasome activation markers) (*Kosmidou et al., 2018*). By contrast, other studies found evidence for inflammasome activation or specifically for NLRP3 inflammasome activation in human RPE cells (*Hollborn et al., 2018*; *Tseng et al., 2013*; *Wang et al., 2019*). However, even if NLRP3 would indeed not be expressed at sufficient levels to be able to activate the inflammasome in the RPE in human eyes with AMD, this would not mean that inflammasome activation does not occur in these cells, as the inflammasome could be activated through other PRRs. Thus, the lack of NLRP3 expression cannot be equated with the lack of inflammasome activation.

Critical unanswered questions are whether NLRP3 inflammasome activation in RPE cells or in non-RPE cells (e.g. macrophages or retinal microglia) promotes AMD pathologies and whether the inflammasome can be activated in AMD not only through NLRP3-dependent but also through NLRP3-independent mechanisms.

Notably, recent *in vitro* studies suggested that sub-lytic effects of the complement pathway activation product C5b-9 (also called membrane attack complex) can induce NLRP3 inflammasome

activation in macrophages (*Laudisi et al., 2013*; *Suresh et al., 2016*; *Triantafilou et al., 2013*). Moreover, the complement activation product C3a has been shown to activate the inflammasome in macrophages *in vitro* as well (*Asgari et al., 2013*). As complement-mediated inflammation occurs in AMD and has been linked to AMD disease progression, this raises the question of whether complement activation products may promote AMD pathologies in part by triggering inflammasome activation in AMD lesions (*Gehrs et al., 2010*). Alternatively, it is possible that complement-mediated inflammation and inflammasome-mediated inflammation are two parallel inflammatory pathways that promote AMD largely independently of each other. Thus, another important unanswered question in the field is whether complement-mediated inflammation promotes AMD pathologies at least in part through inflammasome activation or through inflammasome-independent mechanisms. *Vegfa*$^{hyper}$ mice offer the opportunity to test the roles of complement pathway activation for inflammasome activation and CNV formation, as we observed accumulation of complement C5b-9 in CNV lesions of these mice (*Marneros, 2016*).

Here our data provide important answers to these key questions: (1) we show that NLRP3 inflammasome activation mainly in non-RPE cells and not in RPE cells promotes CNV; (2) we demonstrate that not only NLRP3-dependent but also NLRP3-independent inflammasome activation promotes CNV; (3) we show that the caspase-11 non-canonical inflammasome does not affect CNV; and (4) we provide evidence that complement pathway activation promotes CNV largely independently of inflammasome activation. These findings have significant translational relevance and suggest that combining therapies that target both the canonical inflammasome as well as the complement pathway likely offer synergistic therapeutic benefits for patients with neovascular AMD.

## Results and discussion

### NLRP3 inflammasome activation mainly in non-RPE cells but not in RPE cells promotes CNV

We utilized here a genetic mouse model of neovascular AMD, *Vegfa*$^{hyper}$ mice, which forms spontaneous CNV lesions without artificial injury (*Marneros, 2013*). This model allows us to test effects on both CNV lesion induction (by counting CNV lesion numbers), as well as on CNV lesion growth (measuring CNV lesion area) (*Figure 1A–D*; *Marneros, 2016*). This model is more disease-relevant for AMD than the commonly used laser-injury CNV model, as in the laser model we can only assess CNV lesion size after artificial injury but we cannot assess factors that influence CNV lesion induction and CNV lesion numbers. For all CNV quantifications across all experimental mouse groups, we used 6-weeks-old mice, since at this young age early CNV lesions are present, whereas with progressive age CNV lesions increase in size and adjacent CNV lesions can fuse to form confluent larger lesions (*Figure 1E*; *Marneros, 2013*). Thus, quantifying CNV lesions in young 6-weeks-old age-matched mice allows us to determine effects on early CNV lesion induction and to quantitate accurately the number of initial CNV lesions present at a defined early stage of the disease process before they become confluent larger lesions. Moreover, we already observed in RPE/choroid lysates of 6-weeks-old *Vegfa*$^{hyper}$ mice a strong increase in the inflammasome activation product caspase-1 p10 (p10 levels serve as a direct correlate of inflammasome activity), linking CNV formation with inflammasome activation in these mice already at this young age (*Figure 1F*).

Notably, the size of CNV lesions can be accurately quantified in this AMD mouse model by immunolabeling of choroidal flat mounts with the endothelial cell marker CD31 that strongly labels neovessels that protrude into the sub-RPE space (*Figure 1A–E*). These neovessels affect the overlying RPE morphology, which can be visualized by labeling with phalloidin or immunolabeling for active β-catenin, both outlining RPE cell membranes (*Figure 1A, B, D and E*). Disruption of the typical honeycomb pattern morphology of the RPE occurs in 6-weeks-old *Vegfa*$^{hyper}$ mice only at sites of CD31$^+$ neovessel protrusions into the sub-RPE space (*Figure 1D*). CNV lesions were quantified in this study by co-labeling for CD31 and with phalloidin in choroidal flat mounts and all CNV lesions showed a strict co-occurrence of disruption of RPE morphology with CD31 neovessel protrusions. Thus, all phalloidin-marked CNV lesions analyzed in this study contained CD31$^+$ neovessels.

At sites devoid of CNV lesions, the RPE and retina appear morphologically normal in 6-weeks-old *Vegfa*$^{hyper}$ mice (*Figure 1A and B*; *Marneros, 2013*). By contrast, in aged *Vegfa*$^{hyper}$ mice RPE cells become atrophic and the photoreceptor layer of the retina is attenuated as well, resembling

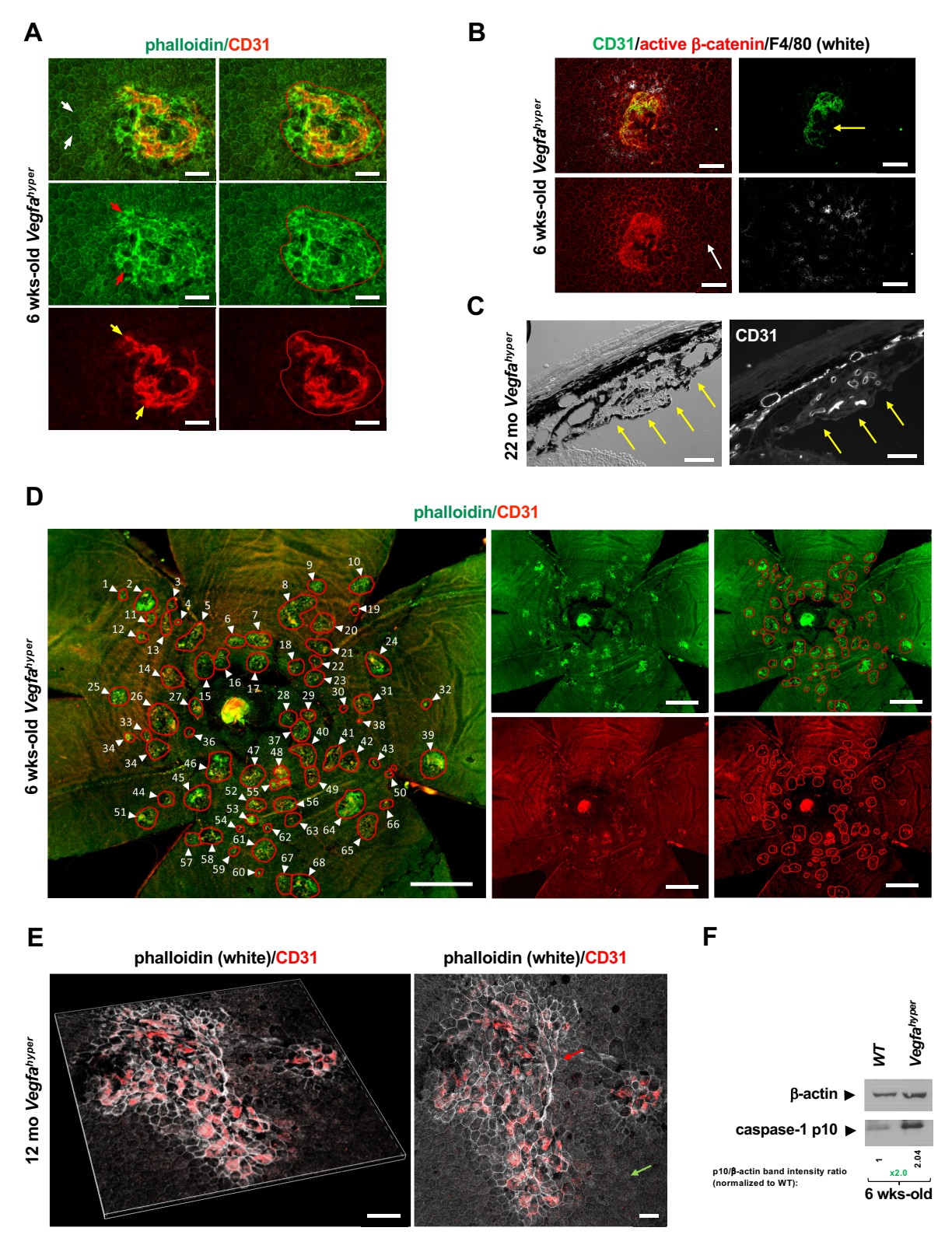

**Figure 1.** Quantifications of CNV lesions in *Vegfa^hyper* mice. (**A**) Disrupted honeycomb pattern morphology of RPE cells and sub-RPE protrusions of CD31⁺ neovessels demarcate a choroidal neovascular lesion in choroidal flat mounts of *Vegfa^hyper* mice. A choroidal flat mount of a *Vegfa^hyper* mouse is shown. Phalloidin staining highlights RPE cell membranes and shows the regular honeycomb pattern of RPE cells at sites devoid of CNV lesions (white arrows). At the site of a CNV lesion, the typical RPE honeycomb pattern is disrupted (phalloidin, green; red arrows). CD31 (red) staining identifies

*Figure 1 continued on next page*

Figure 1 continued

neovessels of a CNV lesion that protrude into the sub-RPE space (yellow arrows). Disruption of regular RPE cell morphology and CD31 staining of neovessels determine the size of the lesion. A polygon is used to quantify the lesion area. Scale bars, 50 µm. 6-weeks-old mouse. (**B**) CNV lesion in a 6-weeks-old *Vegfa*$^{hyper}$ mouse shows that RPE disruption occurs only at sites of choroidal neovascularization due to CD31$^+$ neovessels (green) protruding into the sub-RPE space, whereas RPE cells show a normal honeycomb pattern cellular morphology at sites without CNV lesions (white arrow). Labeling for active β-catenin outlines RPE cell membranes (red), similar to phalloidin staining. F4/80$^+$ cells (white) infiltrate sites of CNV lesions. Pigmented cells cover part of the CNV lesion (yellow arrow). Scale bars, 100 µm. (**C**) Section through a CNV lesion (yellow arrows) in an aged *Vegfa*$^{hyper}$ mouse eye (22-months-old) shows CD31$^+$ neovessels in CNV lesions. Scale bars, 50 µm. (**D**) Representative image of how CNV lesions were quantified in choroidal flat mounts in mice with the *Vegfa*$^{hyper}$ allele. Measurements of multiple lesions in a choroidal flat mount. A choroidal flat mount of a *Vegfa*$^{hyper}$ mouse is shown in which each CNV lesion is counted (numbered, arrows) and each lesion area shown by a polygon. Phalloidin staining (green) outlines CNV lesions and CD31 immunolabeling (red) detects CD31$^+$ neovessels. Increased phalloidin signal at sites of disruption of the regular RPE morphology occurs only at sites of CD31$^+$ neovessels. Scale bars, 500 µm. 6-weeks-old mouse. (**E**) Enlarged confluent CNV lesions can be observed in aged *Vegfa*$^{hyper}$ mice. Even at an advanced age, disruption of the honeycomb pattern RPE morphology is strictly localized to the site of CD31$^+$ neovessel protrusions into the sub-RPE space in CNV lesions (red arrow), whereas RPE areas with no CNV maintain their honeycomb pattern (green arrow). The left image shows Z-stack 3D-rendering of CNV lesions in a 12-months-old *Vegfa*$^{hyper}$ mouse eye and the right image shows the projection of this z-stack. CD31$^+$ neovessels of CNV lesions in red; phalloidin in white. Scale bars, 50 µm. (**F**) RPE/choroid lysates from 6-weeks-old *Vegfa*$^{hyper}$ mice and their WT littermates show that the inflammasome activation product caspase-1 p10 can already be detected at this young age at which CNV lesions were quantified. A > 2 fold increase in p10 levels relative to WT (normalized to β-actin levels) are observed at 6 weeks of age in RPE/choroid lysates of *Vegfa*$^{hyper}$ mice. RPE/choroids of both eyes from four *Vegfa*$^{hyper}$ mice versus four of their WT littermates were pooled. An anti-caspase-1 p10 antibody from Thermo Fisher Scientific was used (PA5-105049) to detect p10.

---

pathologies seen in nonexudative AMD (*Ablonczy et al., 2014*; *Marneros, 2013*; *Marneros, 2016*). However, even in aged *Vegfa*$^{hyper}$ mice co-labeling of choroidal flat mounts for CD31 and with phalloidin shows that the disruption of the honeycomb pattern of RPE morphology co-localizes strictly to the sites of CNV and occurs due to protruding CD31$^+$ neovessels, whereas sites devoid of CNV do not show these RPE changes (*Figure 1E*).

We previously found that inactivation of *Nlrp3* results in reduced CNV lesion numbers in *Vegfa*$^{hyper}$ mice, but we do not know whether this effect is explained by loss of NLRP3 in RPE cells or in non-RPE cells (*Marneros, 2013*; *Marneros, 2016*). Controversy exists particularly about whether NLRP3 is expressed in the RPE in sufficient amounts to be functional and to promote AMD pathologies by activating the inflammasome.

To test whether NLRP3 inflammasome activation in RPE cells versus in non-RPE cells promotes CNV, we established mice that have a constitutively active *Nlrp3* allele only in the RPE, whereas NLRP3 is inactivated in all other cells of these mice (*Figure 2A*). For this purpose, we utilized mice that carry a A350V mutation in the *Nlrp3* gene (*Nlrp3*$^{A350V/A350V}$). This mutation results in uninhibited NLRP3 activation in patients and mice with cryopyrin-associated periodic syndrome (CAPS) (*Brydges et al., 2009*). In these mice a floxed inverted *neoR* cassette (*neoR*$^{fl/fl}$) in intron 2 inhibits *Nlrp3*$^{A350V/A350V}$ expression. Thus, in the absence of Cre, the *neoR* cassette remains in place and prevents expression of *Nlrp3*$^{A350V/A350V}$, rendering the mice null for NLRP3. Only in the presence of Cre recombinase is the *neoR*$^{fl/fl}$ cassette removed, which allows the expression of the mutant *Nlrp3*$^{A350V/A350V}$ that causes constitutive NLRP3 activation. We crossed floxed *neoR*$^{fl/fl}$-*Nlrp3*$^{A350V/A350V}$ mice with *Best1*$^{Cre/+}$ mice (expressing Cre recombinase exclusively in the RPE, also called *VMD2*$^{Cre/+}$) and *Vegfa*$^{hyper}$ mice to generate *Vegfa*$^{hyper}$*Best1*$^{Cre/+}$*Nlrp3*$^{A350V/A350V}$ mice (*Figure 2A*). As Cre recombinase is expressed in *Best1*$^{Cre/+}$ mice only in RPE cells and in no other cell types in the eye (*Figure 2—figure supplement 1*; *He et al., 2014*; *Iacovelli et al., 2011*), the constitutively active *Nlrp3*$^{A350V/A350V}$ mutant allele is exclusively present in the RPE in *Vegfa*$^{hyper}$*Best1*$^{Cre/+}$*Nlrp3*$^{A350V/A350V}$ mice. All other eye tissues (non-RPE cells) of *Vegfa*$^{hyper}$*Best1*$^{Cre/+}$*Nlrp3*$^{A350V/A350V}$ mice do not express Cre and do not remove the *neoR*$^{fl/fl}$ cassette (*Figure 2—figure supplement 1C*), resulting in a lack of *Nlrp3* expression and NLRP3 inflammasome activity in these Cre-negative non-RPE cells (*Brydges et al., 2009*). Therefore, comparing *Vegfa*$^{hyper}$*Best1*$^{Cre/+}$*Nlrp3*$^{A350V/A350V}$ mice with *Vegfa*$^{hyper}$*Nlrp3*$^{-/-}$ mice (which are Cre-negative *Vegfa*$^{hyper}$*Cre*$^{neg}$*Nlrp3*$^{A350V/A350V}$ mice and represent *Vegfa*$^{hyper}$ mice that lack NLRP3 in all cells), allows us to determine whether a constitutively active *Nlrp3* allele selectively in the RPE promotes CNV. Only those choroidal flat mounts of *Vegfa*$^{hyper}$*Best1*$^{Cre/+}$*Nlrp3*$^{A350V/A350V}$ mice were included in CNV quantifications that showed uniform Cre expression in the RPE (*Figure 2A*; *Figure 2—figure supplement 1*).

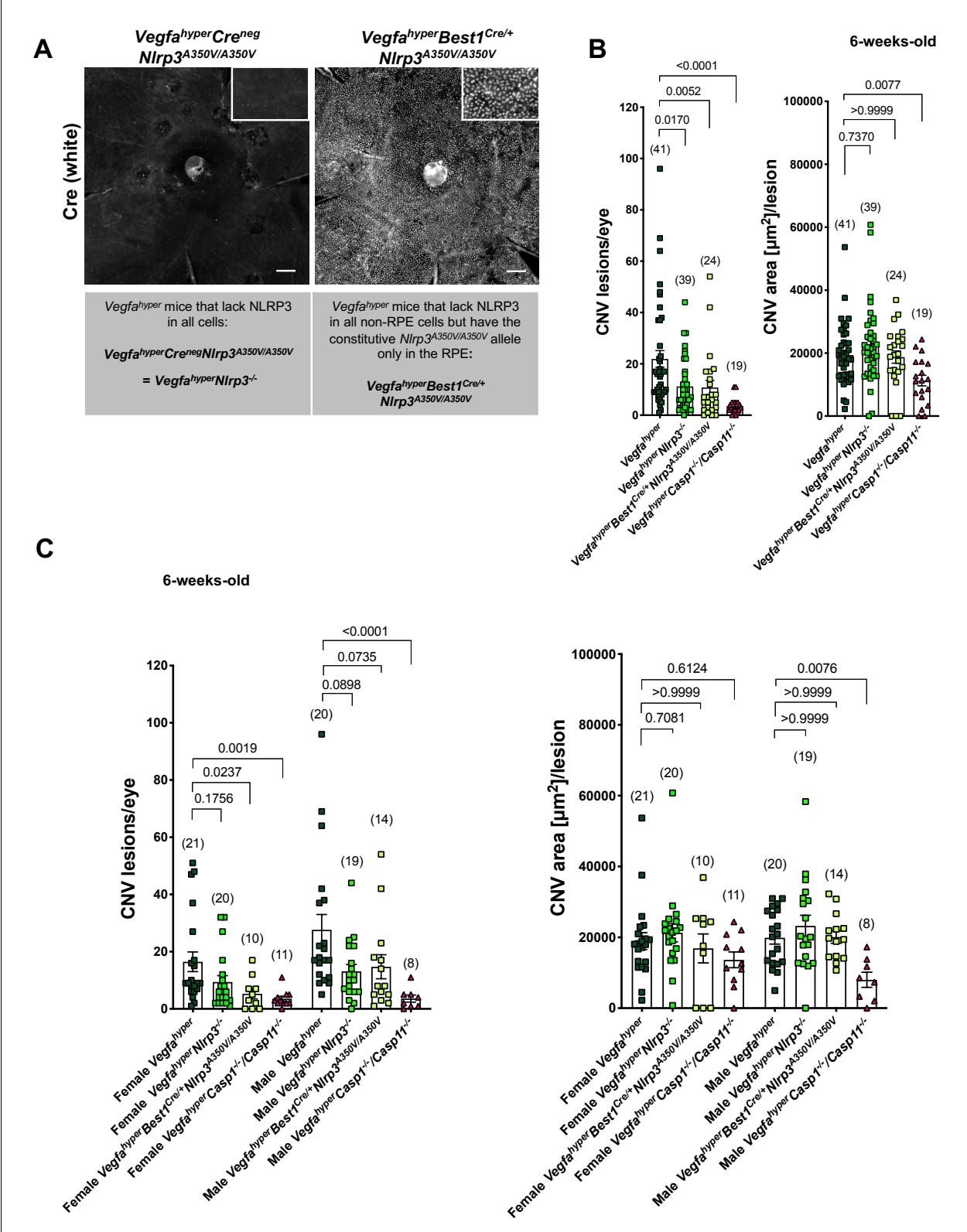

**Figure 2.** NLRP3 inflammasome activation in non-RPE cells but not in RPE cells promotes CNV. (**A**) Nuclear Cre immunostaining (white) in choroidal flat mounts from 6-weeks-old *Vegfa^hyperBest1^Cre/+Nlrp3^A350V/A350V* mice confirms uniform Cre expression in the RPE. Inset shows higher magnification image. No Cre staining is seen in *Vegfa^hyperCre^negNlrp3^A350V/A350V* mice ( = *Vegfa^hyperNlrp3^-/-* mice). Scale bars, 200 μm. (**B-C**). CNV lesion numbers and average CNV lesion area (in μm²/lesion) were measured in 6-weeks-old mice. *Vegfa^hyperNlrp3^-/-* mice (group 2) and *Vegfa^hyperBest1^Cre/+Nlrp3^A350V/A350V*

*Figure 2 continued on next page*

Figure 2 continued

mice (group 3) show significantly decreased CNV lesion numbers compared to *Vegfa^hyper* mice (group 1), but no significant difference in CNV lesion sizes. CNV lesion sizes and numbers between *Vegfa^hyper^Nlrp3^-/-* mice and *Vegfa^hyper^Best1^Cre/+^Nlrp3^A350V/A350V* mice are similar. No marked differences in lesion sizes or lesion numbers were observed among male or female groups of *Vegfa^hyper^Nlrp3^-/-* mice and *Vegfa^hyper^Best1^Cre/+^Nlrp3^A350V/A350V* mice. *Vegfa^hyper^Casp1^-/-^Casp11^-/-* mice (group 4) have fewer CNV lesions than *Vegfa^hyper^Nlrp3^-/-* mice or *Vegfa^hyper^Best1^Cre/+^Nlrp3^A350V/A350V* mice. Values represent total CNV lesion numbers/eye for each mouse and average lesion area/eye for each mouse. Absolute numbers of mice per group are indicated in parentheses. Graphs show mean ± SEM. P-values are shown and were determined by a Kruskal-Wallis test followed by a Dunn's test. The online version of this article includes the following figure supplement(s) for figure 2:

**Figure supplement 1.** RPE-specific Cre expression in *Best1^Cre/+* mice mediates the removal of floxed alleles in the RPE.

Numbers (correlating to lesion induction) and sizes (area, correlating to lesion growth) of CNV lesions were assessed in choroidal flat mounts from 6-weeks-old *Vegfa^hyper* mice (group 1), *Vegfa^hyper^Nlrp3^-/-* mice (with lack of NLRP3 in all cells) (group 2), *Vegfa^hyper^Best1^Cre/+^Nlrp3^A350V/A350V* mice (with the constitutively active *Nlrp3^A350V/A350V* allele in the RPE only while lacking NLRP3 in all non-RPE cells) (group 3), and in *Vegfa^hyper^Casp1^-/-^Casp11^-/-* mice (group 4) that lack all canonical and non-canonical inflammasome activity (*Figure 2B and C*). We previously reported that 6-weeks-old *Vegfa^hyper^Nlrp3^-/-* mice have reduced CNV lesion numbers compared to 6-weeks-old *Vegfa^hyper* mice, but with similar lesion sizes in both groups (*Marneros, 2016*). We show now that the strong reduction in CNV lesion numbers compared to *Vegfa^hyper* mice (group 1) is similar as seen in *Vegfa^hyper^Nlrp3^-/-* mice (group 2) also in *Vegfa^hyper^Best1^Cre/+^Nlrp3^A350V/A350V* mice (group 3), and that CNV lesion sizes are not affected (*Figure 2B and C*). The similar effect on CNV lesion number reduction in *Vegfa^hyper^Best1^Cre/+^Nlrp3^A350V/A350V* mice and *Vegfa^hyper^Nlrp3^-/-* mice shows that the constitutively active *Nlrp3^A350V/A350V* allele in the RPE does not promote CNV lesion formation, whereas NLRP3 inflammasome inactivation in non-RPE cells potently inhibits CNV lesion induction.

## F4/80^+ cells in CNV lesions have high levels of caspase-1 protein

These findings suggest that NLRP3 inflammasome activation particularly in non-RPE cells promotes CNV. As caspase-1 is required for inflammasome activation, detection of caspase-1 or its activation products p10/p20 in a specific cell type suggests that inflammasome activation in these cells is more likely than in cells in which no caspase-1 immunolabeling is detected. Thus, in order to test which cell types show significant levels of caspase-1 protein levels, we performed immunolabeling of choroidal flat mounts and eye sections from *Vegfa^hyper* mice with several different antibodies that detect either caspase-1 or its activation product p10. We observed strong immunolabeling for caspase-1 or its activation product p10 predominantly in F4/80^+ cells in CNV lesions of *Vegfa^hyper* mice, whereas no strong immunolabeling was detected in their RPE cells (*Figure 3A–C*). In the eye, both macrophages as well as retinal microglia express F4/80 (*Yu et al., 2020*). Notably, we observed that a subset of F4/80^+ cells that infiltrate CNV lesions in *Vegfa^hyper* mice and showed caspase-1 p10 labeling also expressed the microglia marker P2RY12, whereas P2RY12^+ microglia in the retina at sites devoid of CNV lesions did not label for caspase-1 p10 (*Figure 3C*; *Figure 3—figure supplement 1*). This suggests that inflammasome activation occurs in both macrophages (F4/80^+P2RY12^neg) and in activated retinal microglia cells (F4/80^+P2RY12^+) that infiltrate CNV lesions in these mice. Together with our observation that NLRP3 inflammasome activation mainly in non-RPE cells promotes CNV, these data suggest that inflammasome activation primarily in F4/80^+ cells and not in RPE cells drives CNV lesion formation in *Vegfa^hyper* mice.

Indeed, F4/80^+ cells infiltrate the subretinal space at the site of RPE barrier breakdown in *Vegfa^hyper* mice and are spatiotemporally associated with activation of retinal glia cells when CNV lesions form at that site (*Marneros, 2013*), suggesting that these cells are important for the initiation of CNV lesion formation. Consistent with this hypothesis, we and others found that ablation of macrophages (by using either *LysM^Cre/+^iDTR* mice treated with diphtheria toxin or using clodronate liposomes) inhibits laser-induced CNV (*Espinosa-Heidmann et al., 2003*; *Marneros, 2013*; *Sakurai et al., 2003*). Thus, the observed strong reduction in CNV lesion formation in *Vegfa^hyper* mice that lack inflammasome activity (*Vegfa^hyper^Casp1^-/-^Casp11^-/-* mice) (*Figure 2B and C*) is likely due to diminished inflammasome-mediated production of proangiogenic cytokines mainly in lesional activated F4/80^+ cells, whereas potential low-level inflammasome activation in RPE cells is less likely

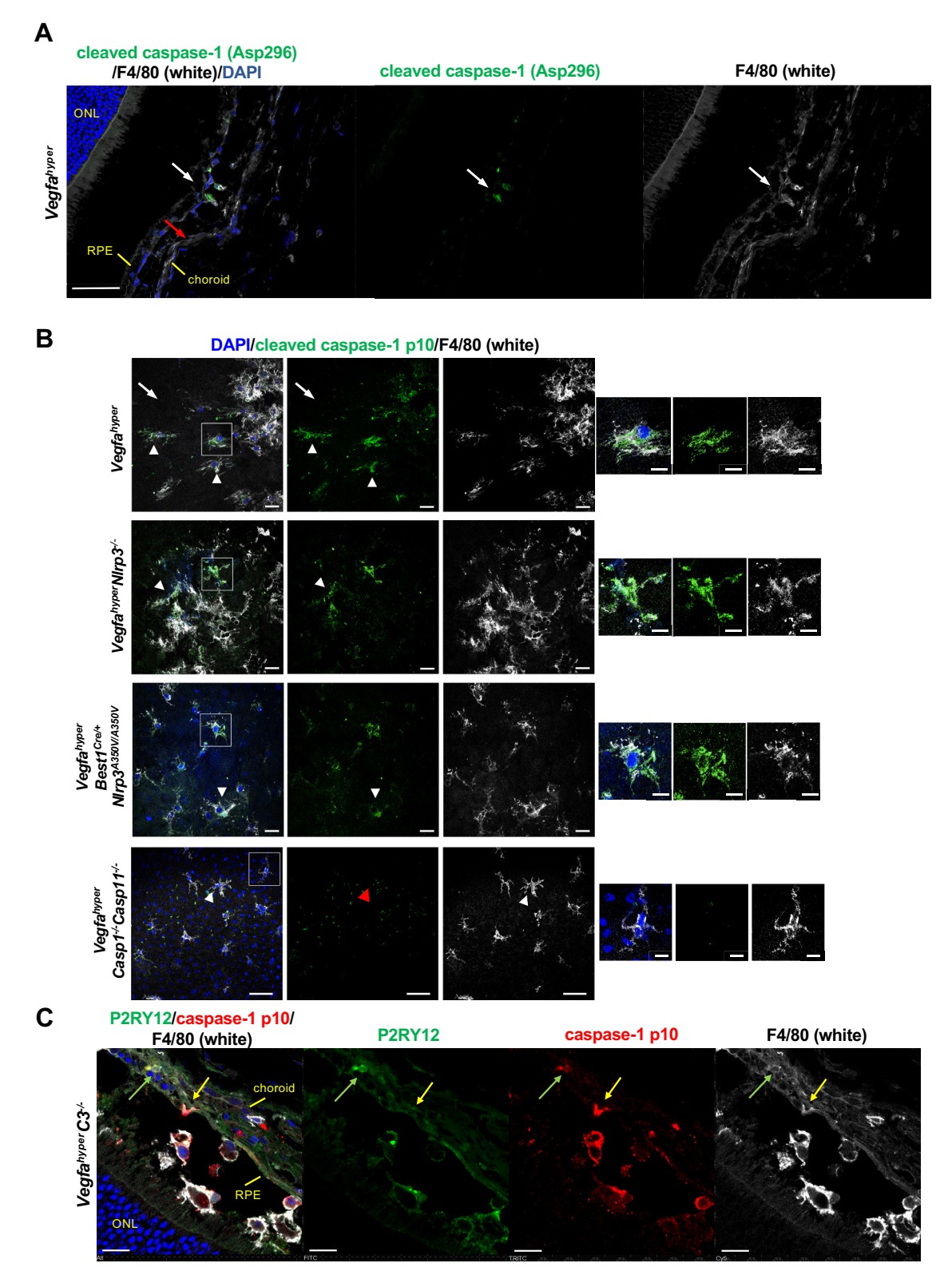

**Figure 3.** F4/80[+] cells in CNV lesions have high levels of caspase-1 protein. (**A**) Immunolabeling of eye sections of *Vegfa^hyper* mice with an antibody that recognizes endogenous levels of caspase-1 protein only when cleaved at Asp296 that occurs with inflammasome activation but does not detect full-length caspase-1 (Cell Signaling Technology Cat# 89332). F4/80[+] cells at the site of a CNV lesion (white) show the presence of cleaved caspase-1 (green, white arrow), whereas this staining is not observed in RPE cells (red arrow) in eyes of *Vegfa^hyper* mice. DAPI nuclear staining in blue. Scale bar, 50

*Figure 3 continued on next page*

*Figure 3 continued*

μm. Location of the outer nuclear layer of the retina (ONL), choroid, and RPE are shown. (**B**) Immunolabeling of choroidal flat mounts for caspase-1 p10 (green) shows predominant staining in F4/80$^+$ cells (white, arrowheads) but not in RPE cells (arrows) in CNV lesions. The antibody (sc-22166, Santa Cruz Biotechnology) detects cleaved caspase-1 p10 (inflammasome activation marker) and was raised against a short amino acid sequence containing the neoepitope at Gly315 of caspase-1 of mouse origin. Proteolytic cleavage of the precursor caspase-1 at Gly315 generates the functional caspase-1 subunits, known as p20 and p10 subunits. Choroidal flat mounts from 3-months-old *Vegfa*$^{hyper}$, *Vegfa*$^{hyper}$*Nlrp3*$^{-/-}$, and *Vegfa*$^{hyper}$*Best1*$^{Cre/+}$*Nlrp3*$^{A350V/}$ $^{A350V}$ mice are shown. No caspase-1 staining is observed even in aged *Vegfa*$^{hyper}$*Casp1*$^{-/-}$*Casp11*$^{-/-}$ mice (red arrowhead), confirming the specificity of the caspase-1 immunolabeling (a representative flat mount of an 18-months-old mouse is shown). DAPI nuclear staining in blue. Right: Higher magnification images of F4/80$^+$ cells (highlighted in boxes of left images). Scale bars, 20 μm (left) and 10 μm (right). (**C**) At sites of CNV lesions in mice with the *Vegfa*$^{hyper}$ allele, a subset of infiltrating F4/80$^+$ cells shows immunolabeling for caspase-1 p10 and is also positive for the microglial marker P2RY12 (green arrows). In contrast, other F4/80$^+$p10$^+$ cells show no staining for P2RY12 (yellow arrows). Representative image of a 12-months-old mouse eye with the *Vegfa*$^{hyper}$ allele is shown (here an eye of a *Vegfa*$^{hyper}$*C3*$^{-/-}$ mouse is shown). Scale bars, 50 μm. DAPI nuclear staining in blue. Location of ONL, choroid, and RPE are shown.

The online version of this article includes the following figure supplement(s) for figure 3:

**Figure supplement 1.** Non-lesional retinal microglia show no caspase-1 p10 immunolabeling.

---

contributing in a significant manner to CNV lesion formation (*Marneros, 2016*). Future approaches to target inflammasome components specifically in macrophages/microglia in this AMD mouse model will help clarify the contributions of these cell types for inflammasome-mediated CNV lesion formation.

## Progressive inflammasome activation in eyes of *Vegfa*$^{hyper}$ mice occurs through both NLRP3-dependent as well as through NLRP3-independent mechanisms

As AMD pathologies and CNV lesions progress and enlarge in an age-dependent manner in *Vegfa*$^{hyper}$ mice (*Figure 1A–E*; *Ablonczy et al., 2014*; *Marneros, 2013*; *Marneros, 2016*), as is the case in patients with neovascular AMD, we tested whether inflammasome activation in RPE/choroid lysates of these mice also increases with age progression and correlates with AMD exacerbation. We found in western blots of RPE/choroid lysates of *Vegfa*$^{hyper}$ mice that an increase in caspase-1 inflammasome activation (increased cleaved caspase-1 p10 levels) compared to WT littermates was present already in young mice (increased ratios of p10/β-actin band intensities in RPE/choroid lysates of 6-weeks-old *Vegfa*$^{hyper}$ mice shown in *Figure 1F* and of 2-months-old *Vegfa*$^{hyper}$ mice shown in *Figure 4A*). Thus, inflammasome activation is already markedly increased in RPE/choroids of young *Vegfa*$^{hyper}$ mice in which we quantitated CNV lesions. Together with our observation that inactivation of inflammasomes in young *Vegfa*$^{hyper}$ mice (*Vegfa*$^{hyper}$*Casp1*$^{-/-}$*Casp11*$^{-/-}$ mice) potently reduced CNV lesion formation, this demonstrates that inflammasome activation already in young *Vegfa*$^{hyper}$ mice promotes CNV lesion formation. The age-dependent exacerbation of AMD pathologies was also associated with a steady further increase in inflammasome activation in RPE/choroids of *Vegfa*$^{hyper}$ mice with age progression and aged *Vegfa*$^{hyper}$ mice showed high levels of caspase-1 p10 in their RPE/choroids (*Figure 4A*). A similar relative increase in caspase-1 p10 levels as observed in RPE/choroids of young *Vegfa*$^{hyper}$ mice compared to their WT controls (2-months-old) was also observed in aged (13-months-old) *Vegfa*$^{hyper}$ mice when compared to their age-matched controls (*Figure 4A*). The specificity of the antibody used to detect caspase-1 and its activation product p10 was confirmed by the absence of a pro-caspase-1 and a caspase-1 p10 band in RPE/choroid lysates from aged *Vegfa*$^{hyper}$*Casp1*$^{-/-}$*Casp11*$^{-/-}$ mice that lack caspase-1 (*Figure 4B*; ratios of p10/β-actin band intensity quantifications shown).

Our finding that inactivation of *Nlrp3* reduces CNV lesion numbers in *Vegfa*$^{hyper}$ mice (*Vegfa*$^{hyper}$*Nlrp3*$^{-/-}$ mice) demonstrates that NLRP3 inflammasome activation promotes CNV lesion formation, but it does not answer the question whether other types of inflammasomes that are activated through different PRRs other than NLRP3 also contribute to CNV lesion formation. Previous studies focused mainly on a role of the NLRP3 inflammasome for AMD pathogenesis but did not examine whether NLRP3-independent inflammasome activation mechanisms may contribute to the progression of AMD pathologies. Thus, we tested whether inflammasome activation in RPE/choroid lysates of *Vegfa*$^{hyper}$ mice can be attributed mainly to NLRP3-mediated inflammasome activation or whether other NLRP3-independent mechanisms can also promote inflammasome activation in

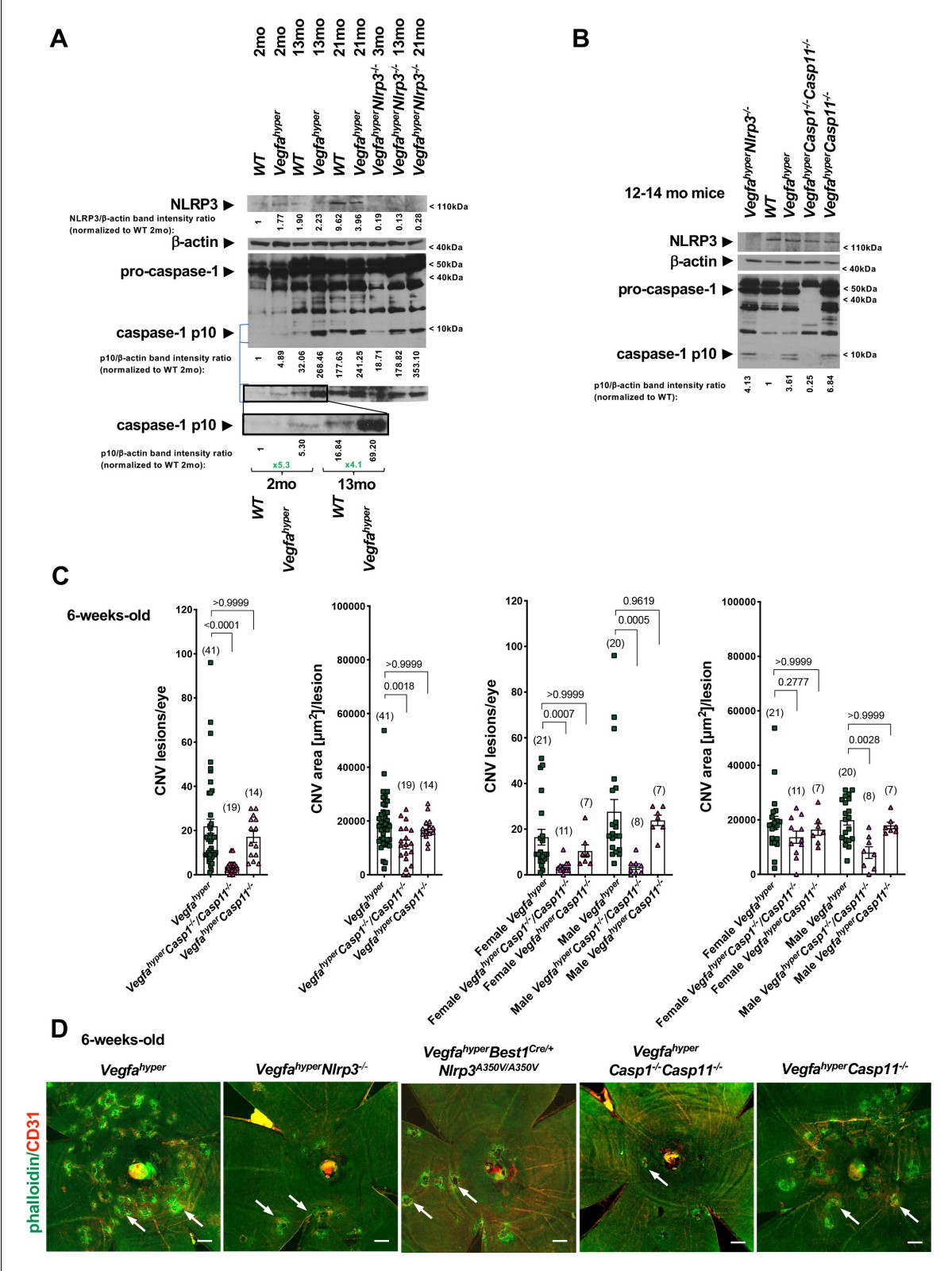

**Figure 4.** Progressive inflammasome activation in eyes of *Vegfa*^hyper^ mice occurs through both NLRP3-dependent as well as through NLRP3-independent mechanisms. (**A**) RPE/choroid lysates of *Vegfa*^hyper^ mice show increased levels of the inflammasome activation marker cleaved caspase-1 p10 compared to age-matched WT mice both at 2 months of age as well as at 13 months of age. Caspase-1 p10 levels increase with progressive age in eyes of both WT and *Vegfa*^hyper^ mice and maintain their relative difference with age. Caspase-1 p10 levels are already markedly increased in RPE/

*Figure 4 continued on next page*

Figure 4 continued

choroid lysates from 2-months-old *Vegfa^hyper* mice when compared to their WT littermates (~5 fold). They are further increased in 13-months-old mice with a similar relative increase in *Vegfa^hyper* mice when compared to their WT littermates (~4 fold). NLRP3 immunoblotting confirms the presence of NLRP3 in RPE/choroid lysates of *Vegfa^hyper* mice already at 2 months of age and shows the lack of NLRP3 protein in *Vegfa^hyper*Nlrp3^-/- mice, validating the specificity of the NLRP3 antibody. Lack of NLRP3 does not prevent inflammasome activation in RPE/choroid lysates and *Vegfa^hyper*Nlrp3^-/- mice show caspase-1 p10 levels in their eyes that increase with progressive age, similarly as observed in *Vegfa^hyper* mice. Four RPE/choroid tissues from four independent mice were pooled for each of the groups indicated. Age of groups are indicated. Ratios of p10/β-actin or NLRP3/β-actin band intensities normalized to 2-months-old WT values are shown. Blue lines indicate p10 bands from the same western blot after re-probing with secondary antibody and longer exposure of the film. Boxed area is magnified and p10 bands are quantified, demonstrating that p10 levels are already increased in 2-months-old *Vegfa^hyper* mouse groups. (B) RPE/choroid lysates of *Vegfa^hyper* mice, *Vegfa^hyper*Nlrp3^-/- mice and *Vegfa^hyper*Casp11^-/- mice show increased levels of caspase-1 p10 compared to those of WT mice. No pro-caspase-1 or p10 were observed in lysates of *Vegfa^hyper*Casp1^-/-Casp11^-/- mice, confirming the specificity of the antibody used (Santa Cruz Biotechnology Cat# sc-514). WT mice, *Vegfa^hyper* mice and *Vegfa^hyper*Casp1^-/-Casp11^-/- mice express NLRP3. Lysates of RPE/choroid tissues from 12 to 14 months-old mice (four RPE/choroids pooled for each group). Ratios of p10/β-actin band intensities normalized to WT values are shown. (C) *Vegfa^hyper*Casp1^-/-Casp11^-/- mice show significantly reduced CNV lesion numbers compared to *Vegfa^hyper* mice. Significantly smaller CNV lesion sizes were observed in male but not in female *Vegfa^hyper*Casp1^-/-Casp11^-/- mice compared to male and female *Vegfa^hyper* mice respectively. Comparing *Vegfa^hyper*Casp11^-/- with *Vegfa^hyper* mice shows no significant difference in either CNV lesion numbers or their sizes. 6-weeks-old mice. Values represent total CNV lesion numbers/eye for each mouse and average lesion area/eye for each mouse. Absolute numbers of mice per group are indicated in parentheses. Graphs show mean ± SEM. P-values are shown and were determined with a Kruskal-Wallis test followed by a Dunn's test. (D) Representative choroidal flat mount images show multiple CD31+ CNV lesions (white arrows) in 6-weeks-old *Vegfa^hyper* and *Vegfa^hyper*Casp11^-/- mice and much fewer lesions in *Vegfa^hyper*Nlrp3^-/-, *Vegfa^hyper*Best1^Cre/+^Nlrp3^A350V/A350V, and *Vegfa^hyper*Casp1^-/-Casp11^-/- mice. Staining for phalloidin in green (outlining CNV lesions prominently) and for CD31 in red. Scale bars, 200 µm. Phalloidin-marked loss of honeycomb morphology of RPE occurs only at sites of CD31+ neovessel protrusion into the sub-RPE space. This can also be seen when assessing red and green channels of these images separately (shown in *Figure 4—figure supplement 1*).

The online version of this article includes the following figure supplement(s) for figure 4:

**Figure supplement 1.** Disruption of honeycomb pattern RPE morphology occurs only at sites of CD31 neovessel protrusions into the sub-RPE space.

CNV. Western blots detected NLRP3 protein already in RPE/choroid lysates from young *WT* and *Vegfa^hyper* mice, whereas NLRP3 protein was absent in RPE/choroid lysates from *Vegfa^hyper*Nlrp3^-/- mice (*Figure 4A and B*). Moreover, RPE/choroid lysates from *Vegfa^hyper*Nlrp3^-/- mice showed the presence of a cleaved caspase-1 p10 band despite the absence of NLRP3 protein, demonstrating that inflammasome activation in eyes of *Vegfa^hyper* mice is not dependent on NLRP3 and can also be mediated through other inflammasome activators/PRRs in the absence of NLRP3 (*Figure 4A and B*). Caspase-1 activation increased in *Vegfa^hyper*Nlrp3^-/- mice in an age-dependent manner similarly as in *Vegfa^hyper* mice (*Figure 4A*). This is also consistent with our observation that strong caspase-1 immunolabeling occurred in F4/80+ cells in CNV lesions of *Vegfa^hyper*Nlrp3^-/- mice to a similar extent as seen in *Vegfa^hyper* mice (*Figure 3B*). Together with the observation of reduced CNV lesion formation in *Vegfa^hyper*Nlrp3^-/- mice compared to *Vegfa^hyper* mice, these data suggest that inflammasome activation in RPE/choroid tissues of *Vegfa^hyper* mice can occur through both NLRP3-dependent and NLRP3-independent mechanisms. These findings are in accordance with our observation that *Vegfa^hyper* mice that lack caspase-1/caspase-11 (and, thus, lack all canonical and non-canonical inflammasomes) exhibit a stronger reduction in CNV lesion numbers compared to *Vegfa^hyper* mice that lack only NLRP3 (*Figure 2B and C*; *Marneros, 2016*). In fact, CNV lesions were fewer and smaller in size (decreased lesion formation as well as a decreased lesion growth) in *Vegfa^hyper*Casp1^-/-Casp11^-/- mice when compared to *Vegfa^hyper*Nlrp3^-/- mice or *Vegfa^hyper*Best1^Cre/+^Nlrp3^A350V/A350V mice (*Figure 2B and C*).

## Non-canonical inflammasome activation does not affect CNV lesions in *Vegfa^hyper* mice

Mice that lack *Casp1* (thus, lacking all canonical inflammasomes as caspase-1 is a required component of all inflammasomes) have also an incidental inactivating mutation in *Casp11* (they are therefore double KOs for both caspase-1 and caspase-11 [referred to as *Vegfa^hyper*Casp1^-/-Casp11^-/-mice; *Kayagaki et al., 2011*]. We show that these *Vegfa^hyper*Casp1^-/-Casp11^-/- mice have indeed no caspase-1 protein [thus, no inflammasome activity], as no pro-caspase-1 nor any caspase-1 p10 can be detected in western blots of RPE/choroid lysates from these mice [*Figure 4B*]). To rule out a significant contribution of caspase-11 to inflammasome activation or CNV lesion formation in eyes of *Vegfa^hyper* mice, we generated *Vegfa^hyper*Casp11^-/- mice, which have

a functional *Casp1* gene (thus, can have canonical inflammasome activation) but lack *Casp11* (*Kayagaki et al., 2011*; *Li et al., 2007*). We show in this western blot that, indeed, lack of caspase-11 does not reduce inflammasome activation (caspase-1 p10 levels are not diminished in RPE/choroid lysates of *Vegfa^hyperCasp11^-/-* mice) (*Figure 4B*). Consistent with this finding, lack of caspase-11 does also not affect CNV lesion numbers or sizes (*Figure 4C and D*; *Figure 4—figure supplement 1*). Thus, caspase-11-independent caspase-1 inflammasome activation that is triggered by both NLRP3-dependent and NLRP3-independent mechanisms contributes to CNV induction in *Vegfa^hyper* mice. Notably, lack of caspase-1/caspase-11 potently reduces not only CNV lesion numbers but also CNV lesion sizes (*Figure 4C*).

Collectively, our findings resolve important open questions in the field by demonstrating that (1) NLRP3 inflammasome activation predominantly in non-RPE cells and not in RPE cells promotes VEGF-A-induced CNV and that (2) NLRP3-independent and caspase-11-independent inflammasome activation mechanisms contribute to CNV lesion formation as well.

## Targeting complement-mediated inflammation inhibits CNV without preventing inflammasome activation

Complement factor H (CFH) has been shown to inhibit CD47-mediated elimination of mononuclear phagocytes that maintains homeostasis in the subretinal space, and AMD-associated CFH variants may through this mechanism cause an accumulation of subretinal mononuclear phagocytes that promote AMD pathologies (*Calippe et al., 2017*). This study provides further evidence that links activated phagocytes to the progression of AMD pathologies. Moreover, several human genetic studies implicated a disease-promoting role of complement-mediated inflammation for neovascular AMD (*Gehrs et al., 2010*). Complement activation pathways converge on the critical complement component C3, leading to the generation of complement activation products C3a, C5a and the membrane attack complex C5b-9, which have been shown to accumulate in AMD lesions in patients (*Johnson et al., 2000*; *Mullins et al., 2014*; *Nozaki et al., 2006*). Indeed, we also observed accumulation of C5b-9 in CNV lesions of *Vegfa^hyper* mice (*Marneros, 2016*), suggesting that complement pathway activation may promote CNV in this neovascular AMD mouse model and that *Vegfa^hyper* mice provide an opportunity to assess the role of complement-mediated inflammation for the manifestation of AMD-like pathologies. Complement C3 is found in the retina particularly along Bruch's membrane and we also observed strong immunolabeling for C3 in CNV lesions of *Vegfa^hyper* mice (*Figure 5A*), whereas no C3 was detected in CNV lesions of *Vegfa^hyperC3^-/-* mice (*Figure 5B*). Consistent with a requirement of C3 for the formation of C5b-9, immunolabeling detected C5b-9 in CNV lesions of *Vegfa^hyper* mice but not in CNV lesions of *Vegfa^hyperC3^-/-* mice (*Figure 5B*; *Figure 5—figure supplement 1*).

Notably, complement activation products C3a and C5b-9 can lead to NLRP3 inflammasome activation in macrophages *in vitro* (*Asgari et al., 2013*; *Brandstetter et al., 2015*; *Laudisi et al., 2013*; *Suresh et al., 2016*; *Triantafilou et al., 2013*). These observations raise the question of whether complement pathway activation promotes CNV through NLRP3 inflammasome activation that leads to the release of pro-inflammatory and proangiogenic IL-1β. However, whether the *in vitro* observation of complement-mediated inflammasome activation also has a role for CNV lesion formation *in vivo* or whether complement affects CNV through inflammasome-independent mechanisms is currently not known. Answering this question has important clinical relevance, as an inflammasome-independent effect of complement-mediated inflammation on CNV progression would suggest that combining therapeutic strategies that target both inflammatory pathways would be expected to be more efficient than when only targeting complement activation or inflammasome activation alone. We hypothesize that several factors that accumulate at sites of evolving CNV lesions in *Vegfa^hyper* mice and that have been linked to inflammasome activation (such as increased oxidative stress, sub-RPE lipid deposits, and complement activation products) may all promote inflammasome activation and, thereby, increase CNV formation. However, whether complement pathway activation products contribute in a significant way to inflammasome activation at sites of CNV lesion formation in *Vegfa^hyper* mice is not known.

Thus, we next tested the effects of inactivation of complement C3 on inflammasome activation and the manifestation of neovascular AMD-like pathologies in *Vegfa^hyper* mice by generating *Vegfa^hyper* mice that lack complement C3 (*Vegfa^hyperC3^-/-* mice) (*Figure 5B*; *Wessels et al., 1995*). We found that 6-weeks-old *Vegfa^hyperC3^-/-* mice had significantly fewer CNV lesions than age-matched

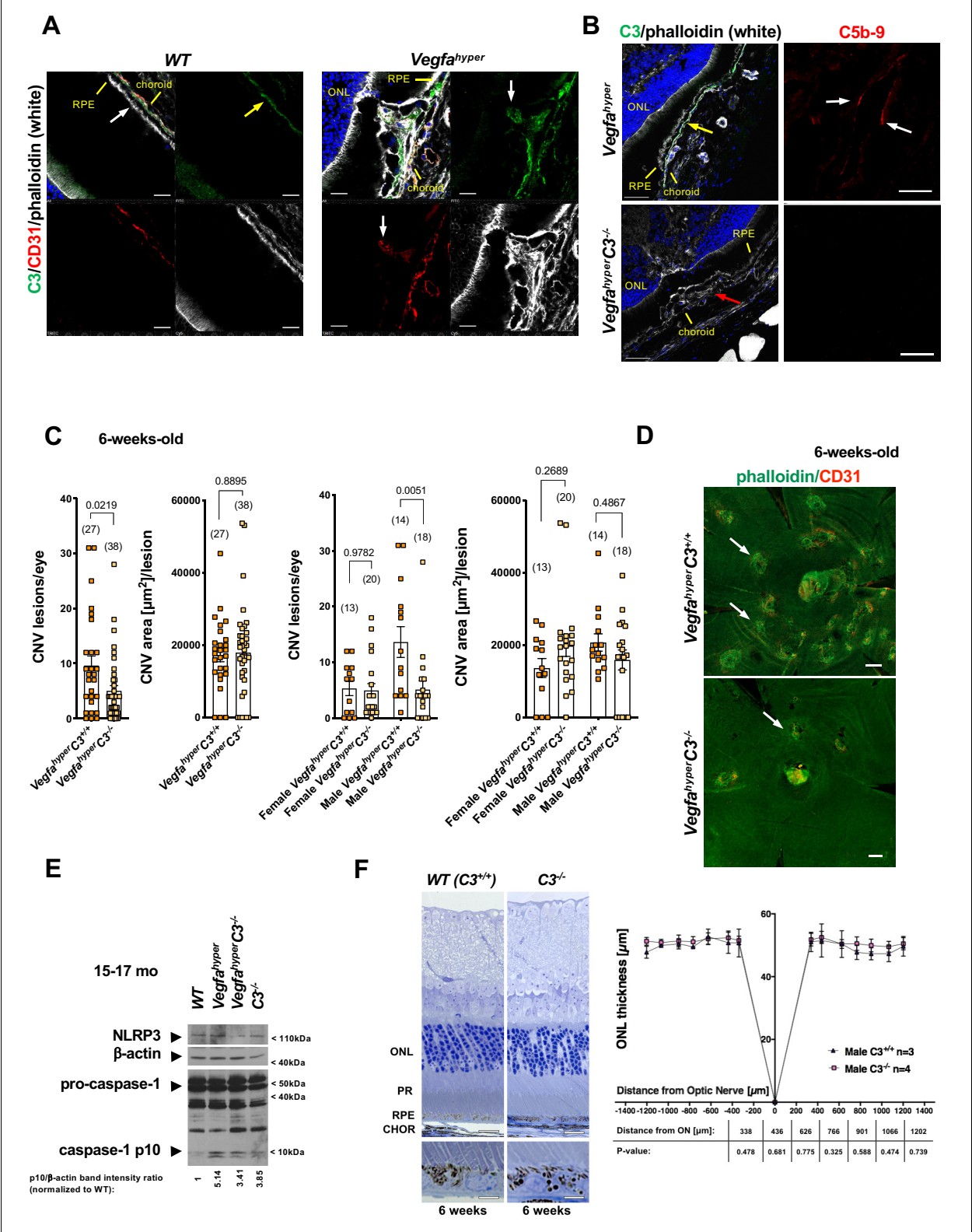

**Figure 5.** Targeting complement-mediated inflammation inhibits CNV without preventing inflammasome activation. (**A**) Left: Complement C3 (yellow arrow, green) is localized particularly along Bruch's membrane under RPE cells (white arrow) in *WT* mice. Right: Strong C3 immunolabeling (green) is also observed in CNV lesions of *Vegfa^hyper* mice, where it co-localizes with CD31⁺ neovessel protrusions (red; white arrows). 12-months-old mice. Scale bars, 50 μm. Location of ONL, choroid, and RPE are shown. (**B**) Left: C3 (green, yellow arrow) is detected in CNV lesions in eyes of *Vegfa^hyper* mice but

*Figure 5 continued on next page*

*Figure 5 continued*

not in CNV lesions in *Vegfa^hyper^C3^-/-* mice (CNV lesion shown by a red arrow), confirming loss of C3 protein in *C3^-/-* mice. Right: CNV lesions in *Vegfa^hyper^* mice show immunolabeling for C5b-9 (red, white arrows), whereas no C5b-9 immunolabeling is detected in CNV lesions in *Vegfa^hyper^C3^-/-* mice. 12-months-old mice. Scale bars, 50 μm (left), 20 μm (right). C5b-9 immunolabeling (red) from co-immunolabeling image shown in *Figure 5— figure supplement 1*. Locations of ONL, choroid, and RPE are shown. (C) *Vegfa^hyper^C3^-/-* mice show significantly fewer CNV lesions compared to littermate *Vegfa^hyper^* mice (*Vegfa^hyper^C3^+/+* mice), but this difference was only observed in male mice. Male *Vegfa^hyper^* mice have more CNV lesions compared to females, whereas there was no marked difference in lesion numbers for male and female *Vegfa^hyper^C3^-/-* mice. Inactivation of C3 did not significantly affect CNV lesion sizes. 6-weeks-old mice. Values represent total CNV lesion numbers/eye for each mouse and average lesion area/eye for each mouse. Absolute numbers of mice per group are indicated in parenthesis. Graphs show mean ± SEM. P-values are shown (two-tailed) and were determined with a Mann-Whitney test. (D) Representative choroidal flat mount images of 6-weeks-old male mice show multifocal CD31^+ CNV lesions (white arrows) in *Vegfa^hyper^* mice and significantly fewer CNV lesions in *Vegfa^hyper^C3^-/-* mice. Staining with phalloidin in green and immunolabeling for CD31 in red. Scale bars, 200 μm. (E) Lack of complement C3 does not prevent inflammasome activation in eyes of *Vegfa^hyper^C3^-/-* mice. Two RPE/choroid tissue lysates were pooled from 15 to 17 months-old mice. Ratios of p10/β-actin band intensities normalized to *WT* values are shown. (F) C3 deficiency does not cause significant retinal morphological defects at 6 weeks of age. Representative sections through posterior eyes of 6-weeks-old male *WT* mice and *C3^-/-* mice are shown. Scale bars, top 20 μm, bottom 5 μm. Outer nuclear layer (ONL) thickness was measured at seven distances along the meridian in eyes of 6-weeks-old mice. Values represent average ONL thickness in μm per group. SEMs are shown for each mean. Student's t-tests were performed for values measured at equal distances from the optic nerve (ON). No significant difference in ONL thickness between *WT* mice and *C3^-/-* mice was observed. PR: photoreceptors; CHOR: choroid.

The online version of this article includes the following figure supplement(s) for figure 5:

**Figure supplement 1.** Complement C5b-9 is observed in CNV lesions of *Vegfa^hyper^* mice, whereas no C5b-9 is observed in CNV lesions of *Vegfa^hyper^C3^-/-* mice.

**Figure supplement 2.** *C3^-/-* mice maintain the typical honeycomb pattern RPE morphology even at 12 months of age.

---

*Vegfa^hyper^* littermates (corresponding to *Vegfa^hyper^C3^+/+* mice without inactivation of *C3*), but no difference in CNV lesion size was observed (*Figure 5C and D*). Notably, female *Vegfa^hyper^* mice tend to have fewer CNV lesions than male *Vegfa^hyper^* mice (*Figures 2C* and *4C*). However, the potent relative reduction in VEGF-A-induced CNV lesion numbers in mice lacking C3 compared to their *Vegfa^hyper^* littermates was observed only in male mice but not in female mice (*Figure 5C*). This is consistent with the previously reported observation that female C57bl6 mice show a reduced ability to promote inflammation via C5b-9, while they show normal complement cascade functionality at the level of C3 (*Kotimaa et al., 2016*). Thus, the difference in CNV lesion numbers between male and female *Vegfa^hyper^C3^-/-* mice also suggests that VEGF-A-induced CNV lesion formation in *Vegfa^hyper^* mice is likely promoted by terminal complement pathway activation.

Importantly, we observed a > 3 fold increase in caspase-1 p10 levels in RPE/choroid lysates of *Vegfa^hyper^C3^-/-* mice (lacking complement C3) compared to age-matched WT mice, demonstrating that lack of C3 does not prevent inflammasome activation in these eyes (*Figure 5E*; ratios of p10/β-actin band intensity quantifications shown). A moderate decrease in p10 levels in RPE/choroid lysates of *Vegfa^hyper^C3^-/-* mice compared to those of age-matched *Vegfa^hyper^* mice may indicate a minor contribution of complement pathway activation to inflammasome activation in RPE/choroids. However, the findings show that C3 is not required for inflammasome activation in RPE/choroid lysates of *Vegfa^hyper^* mice and that inflammasome activation can occur through C3-independent parallel activation mechanisms. We confirmed this also by immunolabeling for caspase-1 p10 in eyes of *Vegfa^hyper^C3^-/-* mice, where we observed p10 staining in F4/80^+ activated macrophages and retinal microglia that infiltrate CNV lesions (*Figure 3C*). As lack of C3 results in fewer CNV lesions in male *Vegfa^hyper^C3^-/-* mice compared to male *Vegfa^hyper^* mice in a highly significant manner despite inflammasome activation in their RPE/choroids, this suggests that C3 affects CNV lesion formation mainly through inflammasome-independent mechanisms. This conclusion is also supported by the data showing that *Vegfa^hyper^* mice lacking all inflammasomes (*Vegfa^hyper^Casp1^-/-^Casp11^-/-* mice) have fewer CNV lesions than *Vegfa^hyper^C3^-/-* mice (if C3 would be critical for inflammasome activation, then *Vegfa^hyper^C3^-/-* mice should have a similarly strong reduction in CNV lesions as *Vegfa^hyper^Casp1^-/-^Casp11^-/-* mice) (*Figure 6*).

Notably, the observation that CNV lesion induction (CNV numbers) is promoted by complement C3, but not their growth (CNV area) (*Figure 5C*), was also made when targeting NLRP3 (*Figure 2B and C*). In contrast, when targeting all inflammasomes (in *Vegfa^hyper^Casp1^-/-^Casp11^-/-* mice), we observed not only a much more potent inhibition of CNV lesion numbers but also of their growth

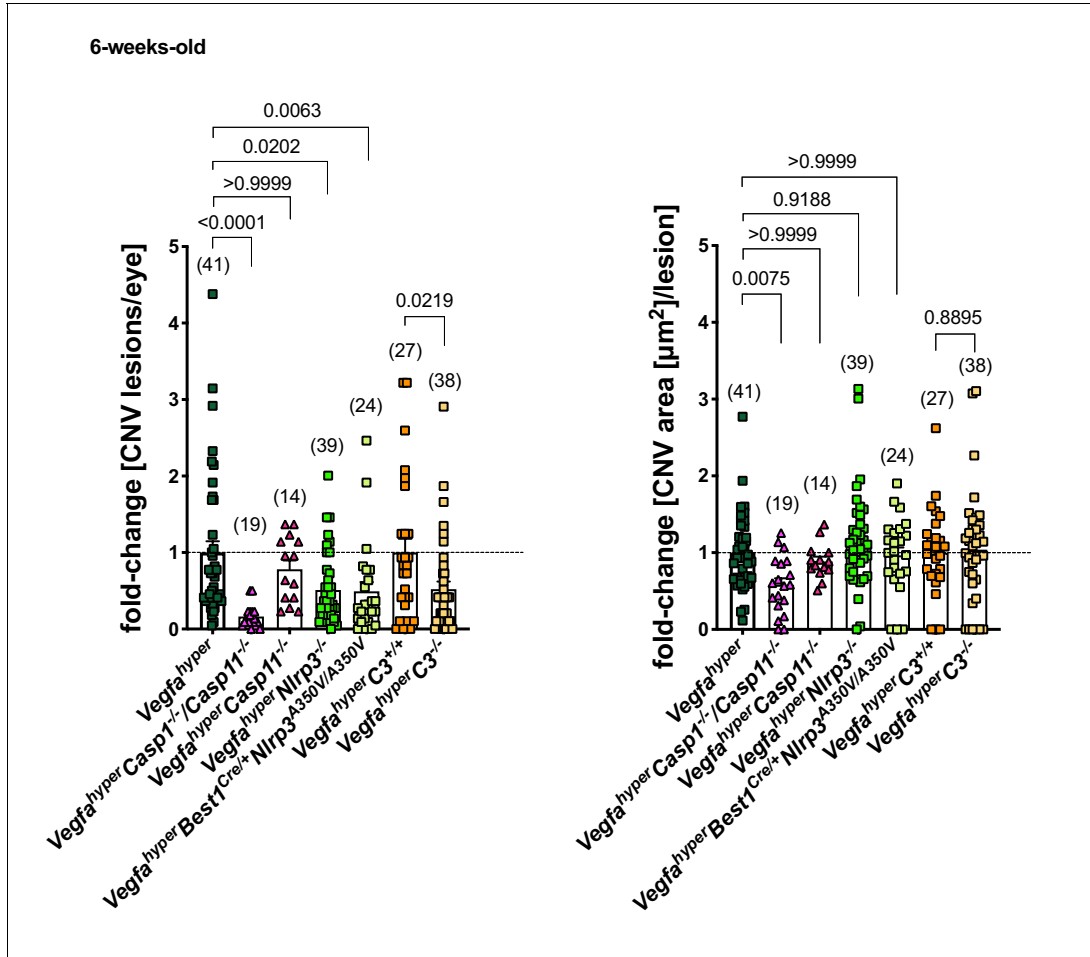

**Figure 6.** Inactivation of all inflammasomes inhibits CNV lesion formation significantly more than when targeting only NLRP3 or C3. Comparison of fold-changes in CNV lesion numbers and sizes in all strains relative to *Vegfa^hyper* mice. 6-weeks-old mice. *Vegfa^hyperC3^{+/+}* mice are littermates of *Vegfa^hyperC3^{-/-}* mice that are WT for C3. Graphs show mean ± SEM. Absolute numbers of mice per group are indicated in parentheses. P-values are shown. Kruskal-Wallis test followed by Dunn's test.

(CNV area) (*Figure 2B and C*). This suggests that CNV lesion growth is at least in part also affected by NLRP3- and C3-independent mechanisms.

C3 inactivation caused no major morphological abnormalities of the RPE or retina when examined at 6 weeks of age, as assessed by cross-sectional morphology of the RPE and the outer nuclear layer (ONL) as well as by RPE staining of choroidal flat mounts with phalloidin (*Figure 5D and F*). Choroidal flat mounts of 12-months-old *C3^{-/-}* mice showed that RPE cells maintain their typical honeycomb pattern morphology even with progressive age (*Figure 5—figure supplement 2*). Notably, previously published detailed studies showed that aged *C3^{-/-}* mice developed retinal abnormalities that were not observed in 6-weeks-old mice, confirming our observation that at the time in which we quantified CNV lesions (6 weeks of age) no RPE/retinal abnormalities could be attributed to C3 deficiency (*Hoh Kam et al., 2013*; *Mukai et al., 2018*; *Yu et al., 2012*). While these studies show that complement C3 plays a role in the maintenance of retinal function with age progression, it remains to be shown whether these retinal abnormalities in aged *C3^{-/-}* mice are a consequence of life-long C3 deficiency and whether targeting C3 or other complement pathway components in aged adults impairs retinal function. Despite the observation of retinal abnormalities in aged *C3^{-/-}* mice that completely lack complement activity, it is possible that strategies to only attenuate complement-mediated inflammation in aged adults with AMD may have therapeutic benefits while maintaining low-level complement pathway activity to not cause detrimental effects on retinal homeostasis.

Collectively, our data demonstrate that VEGF-A-induced neovascular AMD-like pathologies are promoted through inflammasome activation that occurs predominantly in non-RPE cells, and which is only in part mediated by NLRP3 but can also occur independently of NLRP3, caspase-11, or complement C3 (*Figure 6*). Thus, therapeutic strategies that combine approaches to inhibit both inflammasome activation and complement-mediated inflammation could result in more potent inhibition of neovascular AMD than either treatment approach alone.

## Materials and methods

### Animals

For all animal studies, institutional approval by Massachusetts General Hospital was granted. Experiments were performed in compliance with ARRIVAL guidelines. The generation of *Vegfa^hyper* mice was previously reported (*Miquerol et al., 1999*). The increase in VEGF-A levels occurs in these mice as a consequence of the heterozygous insertion of an IRES-NLS-lacZ-SV40pA sequence +202 bp 3′ to the STOP codon into the 3′-UTR of the *Vegfa* gene locus (*Miquerol et al., 1999*). The observed ocular abnormalities always co-segregated with the *Vegfa^hyper* allele (*Vegfa^lacZ/WT*) and were not observed in any of the examined control littermate mice (*Ablonczy et al., 2014*; *Marneros, 2013*; *Marneros, 2016*). *Vegfa^hyper* mice were crossed with *C3^-/-*, *Casp1^-/-/Casp11^-/-*, *Casp11^-/-*, and *Nlrp3^A350VneoR* mice (mice were obtained from JAX laboratories, except *Casp1^-/-/Casp11^-/-* that were obtained from the original investigator) (*Brydges et al., 2009*; *Hara et al., 2006*; *Li et al., 2007*; *Li et al., 1995*; *Wessels et al., 1995*). Of note, mice lacking *Casp1* have an incidental *Casp11* mutation (*Casp1* and *Casp11* are too close in the genome to be segregated by recombination), rendering them double KOs for both caspase-1 and caspase-11 (*Kayagaki et al., 2011*). In contrast, *Casp11* null mice have a functional *Casp1* gene. *Casp1^-/-/Casp11^-/-* mice and *Casp11^-/-* mice have both been shown to lack functional caspase-11 protein (*Kayagaki et al., 2011*). Homozygous *Nlrp3^A350V* mice contain a floxed *neo* cassette in reverse orientation (*neoR*) in intron 2 leading to the lack of functional *Nlrp3* gene expression in the absence of Cre expression (here referred to as *Nlrp3^-/-* mice) (*Brydges et al., 2009*). *Vegfa^hyper Nlrp3^A350V/A350V* mice were crossed with *Best1^Cre/+* mice (*Iacovelli et al., 2011*). The specificity of Cre recombinase expression in *Best1^Cre/+* mice has been extensively analyzed previously, and it was shown both in eye sections as well as in choroidal flat mounts that Cre recombinase expression in the eyes of these mice occurs specifically in RPE cells and not in other cells of the eye (*Iacovelli et al., 2011*), which we confirmed in our previous study as well (*He et al., 2014*). Moreover, we and others showed in these previous publications that co-labeling for phalloidin (outlining the typical honeycomb morphology of RPE cells) and with an antibody against Cre recombinase provides an accurate visualization of the numbers of RPE cells that express Cre recombinase. We confirmed these results in extensive studies by combining Cre immunolabeling with markers that accurately visualize RPE cells (e.g. phalloidin staining or active β-catenin immunolabeling, both highlighting RPE cell membranes) (*Figure 2A*; *Figure 2—figure supplement 1*; *He et al., 2014*). Only those choroidal flat mounts of *Vegfa^hyper Best1^Cre/+ Nlrp3^A350V/A350V* mice were included in CNV quantifications that showed uniform Cre expression in the RPE (*Figure 2A*; *Figure 2—figure supplement 1A and B*). We also confirmed that in *Vegfa^hyper Best1^Cre/+ Nlrp3^A350V/A350V* mice *Best1^Cre/+*-mediated removal of the floxed *neoR* cassette in intron 2 of the *Nlrp3* gene indeed occurs only in RPE cells. RPE cells were isolated from *Vegfa^hyper Best1^Cre/+ Nlrp3^A350V/A350V* mice with dispase-II treatment (20 mg/ml) (D4693, Sigma) and PCR of isolated genomic DNA with primers flanking the floxed *neoR* cassette in intron 2 of the *Nlrp3* gene showed the presence of a PCR product without the *neoR* sequence (*Nlrp3^A350V* allele from the subset of RPE cells in which Cre was active) in addition to a PCR product containing the *neoR* cassette (*Nlrp3^-/-* allele from the subset of RPE cells in which Cre was not active). In contrast, RPE cells from *Vegfa^hyper Nlrp3^-/-* mice without the Cre showed only the PCR product containing the *neoR* cassette (*Nlrp3^-/-* allele). PCR primers used were: 5′-GCTACTTCCATTTGTCACGTCC-3′ (mutant forward), 5′-CACCCTGCATTTTGTTGTTG-3′ (WT forward), 5′-CGTGTAGCGACTGTTGAGGT-3′ (common). Thus, Cre is not only expressed in the RPE of *Vegfa^hyper Best1^Cre/+ Nlrp3^A350V/A350V* mice but also active and mediates Cre-mediated excision of floxed alleles that lead to the *Nlrp3^A350V/A350V* mutant alleles (*Figure 2—figure supplement 1C*). Notably, no Cre-mediated excision of the floxed *neoR* cassette was observed in the retina, lens, or choroid in eyes of *Vegfa^hyper Best1^Cre/+ Nlrp3^A350V/A350V* mice.

Lack of caspase-1 and NLRP3 protein in $Casp1^{-/-}/Casp11^{-/-}$ mice and $Nlrp3^{-/-}$ mice respectively were confirmed in western blotting experiments of RPE/choroid lysates of these mice with validated anti-caspase-1 or anti-NLRP3 antibodies (*Figure 4A and B*). Lack of C3 protein in eyes of $C3^{-/-}$ mice was confirmed by immunolabeling with anti-C3 antibodies (*Figure 5B*). All mice were housed under conventional breeding conditions.

## Morphological examination of eyes

For semithin sections, eyes were fixed in 1.25% paraformaldehyde and 2.5% glutaraldehyde in 0.1 M cacodylate buffer (pH 7.4). After post-fixation in 4% osmium tetroxide, and dehydration steps, tissues were embedded in TAAB epon (Marivac Ltd.). 1 μm-thin sections were used for toluidine blue staining and light microscopy. Outer nuclear layer average thickness of 3 different slides at distances of 338.5, 436.5, 626, 766.5, 901.5, 1066, 1202.5 μm from the optic nerve was determined for each eye using Zeiss Zen blue 2.0 software. Genotype information was concealed when ONL thickness was determined. Results were subsequently assigned to genotypes and differences between mouse strains were determined. P-values were calculated with a two-tailed unpaired Student's t-test. Representative images were obtained from each eye.

## Western blotting

Freshly dissected RPE/choroid tissues were lysed in NP-40 lysis buffer (Life Technologies) with 1 mM PMSF and protease inhibitor cocktail using the Qiagen TissueLyser II. After centrifugation, the supernatant was used for protein quantification. RPE/choroid tissues from WT mice, $Vegfa^{hyper}$ mice, $Vegfa^{hyper}Nlrp3^{-/-}$ mice, $Vegfa^{hyper}Best1^{Cre/+}Nlrp3^{A350V/A350V}$ mice, $Vegfa^{hyper}Casp1^{-/-}Casp11^{-/-}$ mice, $Vegfa^{hyper}Casp11^{-/-}$ mice, $C3^{-/-}$ mice, and $Vegfa^{hyper}C3^{-/-}$ mice were used for western blotting. Equal protein loading was assessed using a polyclonal rabbit anti-mouse β-actin antibody (Lab Vision Cat# RB-9421-P0, RRID:AB_720055) at a dilution of 1:1000. NLRP3 was detected using a monoclonal rabbit anti-mouse NLRP3 antibody (Cell Signaling Technology Cat# 15101, RRID:AB_2722591) at a dilution of 1:1000 (*Kosmidou et al., 2018*). For detection of pro-caspase-1 and activated caspase-1 p10, a polyclonal rabbit anti-mouse antibody (Santa Cruz Biotechnology Cat# sc-514, RRID:AB_2068895) was used at a dilution of 1:200. The specificity of the NLRP3 and caspase-1 p10 antibodies was confirmed by the absence of a signal in RPE/choroid lysates from $Vegfa^{hyper}$ mice lacking either NLRP3 or caspase-1. An anti-caspase-1 p10 antibody from Thermo Fisher Scientific (Thermo Fisher Scientific Cat# PA5-105049, RRID:AB_2816522) was used as well. HRP-conjugated secondary antibodies were used and chemiluminescence signal was determined with SuperSignal WestPico chemiluminescent substrate (Thermo Fisher, Waltham, MA). p10, NLRP3, and β-actin band intensities were quantified with Image J.

## Immunolabeling and choroidal flat mounting

CNV lesions in mice with the $Vegfa^{hyper}$ allele can be accurately quantitated in choroidal flat mounts by co-labeling with phalloidin and immunolabeling for the endothelial cell marker CD31: (1) neovessels in CNV lesions show strongly increased CD31 labeling (allowing distinction from quiescent normal vessels); (2) phalloidin labeling can discern normal RPE cells by highlighting their cell membranes that show a typical honeycomb pattern, whereas this RPE cell morphology is disrupted at the site of CNV lesions by protruding CD31$^+$ neovessels (*Ablonczy et al., 2014*; *Marneros, 2013*; *Marneros, 2016*). Thus, co-labeling for CD31 and with phalloidin provides a sharp demarcation of CNV lesions from surrounding tissue on choroidal flat mounts, allowing for precise measurements of CNV lesion numbers and CNV lesion areas (*Figure 1* shows how CNV quantifications were performed).

The following protocol was used for choroidal flat mount stainings. Whole eyes were enucleated and fixed in 4% paraformaldehyde overnight at 4°C and then washed in PBS. For choroidal flat mounts anterior segment, lens and retina were removed. RPE/choroid tissues were permeabilized in 0.5% Triton X-100 and blocked with 5% serum in which the secondary antibodies were raised for 30 min. Incubation with primary antibodies in the blocking solution was performed overnight at 4°C. Tissues were then washed in PBS. The following primary antibodies were used at a 1:50 dilution: rat anti-mouse CD31 (BD Biosciences Cat# 550274, RRID:AB_393571) , Alexa-647-conjugated rat anti-mouse F4/80 (BioLegend Cat# 123101, RRID:AB_893504), mouse anti-Cre (Millipore Cat# MAB3120,

RRID:AB_2085748), and goat anti-cleaved-caspase-1 p10 (Santa Cruz Biotechnology Cat# sc-22166, RRID:AB_2068884). The sc-22166 antibody from Santa Cruz against cleaved caspase-1 p10 is a goat polyclonal antibody raised against a short amino acid sequence containing the neoepitope at Gly315 of caspase-1 of mouse origin. Proteolytic cleavage of the precursor caspase-1 at glycine residue 317 in humans and 315 in mouse generates the functional caspase-1 subunits, known as p20 and p10 subunits. The specificity of the immunolabeling with this antibody to detect caspase-1-specific peptides was confirmed by the absence of staining in choroidal flat mounts of $Vegfa^{hyper}Casp1^{-/-}Casp11^{-/-}$ mice that lack caspase-1. Eye sections were also stained with an antibody that recognizes endogenous levels of caspase-1 protein only when cleaved at Asp296 but that does not cross-react with full-length caspase-1 (Cell Signaling Technology Cat# 89332). A monoclonal antibody that detects P2RY12 was obtained from BioLegend (BioLegend Cat# 848001, RRID:AB_2650633).

Cytoskeletal staining was performed with Alexa-488- or Alexa-647-conjugated phalloidin (Life Technologies), which outlines the honeycomb pattern of RPE cells. In addition, RPE cell membranes were identified by immunolabeling for active β-catenin (Cell Signaling Technology Cat# 8814, RRID: AB_11127203). For C3 immunolabeling of eye sections, we used a rabbit polyclonal anti-C3 antibody, whose lack of staining in $C3^{-/-}$ mice confirmed the specificity of this antibody (Abcam Cat# ab11887, RRID:AB_298669). Eye sections were also labeled with rabbit polyclonal anti-C5b-9 antibodies (Abcam Cat# ab58811, RRID:AB_879748). Co-labeling experiments were combined with single-labeling experiments and experiments omitting both either the primary or the secondary antibodies to distinguish immunolabeling from autofluorescence. Secondary Alexa-488, Alexa-555, or Alexa-647 antibodies (at a 1:100 dilution) (Life Technologies) were incubated for 3 hr at room temperature in the dark. DAPI (Life Technologies) was used for the staining of nuclei. Subsequently to immunolabeling, RPE/choroid tissues were radially cut eight times and mounted on a slide for microscopic analysis. For immunolabeling of eye sections, frozen OCT-embedded tissue blocks were cut at 7μm thickness.

## Quantification of the size and number of CNV lesions

Allocation, treatment, and handling of mice were the same across all study groups. For all mice randomization was performed and data were collected and processed randomly. Investigators quantifying CNV lesions were blinded to the genotype of the mice. We used 6-weeks-old mice for all CNV quantifications in order to have a uniform age-controlled population of mice in which early stages of CNV lesions are present. Immunolabeling was performed with anti-CD31 antibodies to detect blood vessels and combined with Alexa-488-conjugated phalloidin labeling. Anti-Cre antibody was used to assess the extent of Cre$^+$ RPE in $Vegfa^{hyper}Best1^{Cre/+}Nlrp3^{A350V/A350V}$ mice. Neovessels in CNV lesions show strong CD31 immunoreactivity allowing for a clear separation of CNV lesions from surrounding tissues (*Figure 1*). Similarly, phalloidin-Alexa488 labeling outlines the typical honeycomb pattern of RPE cells. At sites of CNV lesions, this pattern is disrupted by protruding CD31$^+$ neovessels, providing an accurate demarcation of CNV lesion areas (*Figure 1*). The area of each CNV lesion was measured in μm$^2$ using Zen blue 2.0 software. The number of CNV lesions and the average CNV lesion area for each eye was determined. Genotyping information was concealed when lesion sizes and numbers were determined to ensure unbiased analyses. Results were subsequently assigned to genotypes and differences between mouse strains were determined. CNV lesion number and average CNV area per mouse are shown (GraphPad Prism version 8.4.3). Sample size requirements were determined based on our previous CNV quantifications in $Vegfa^{hyper}$ mice and those lacking inflammasome components (*Marneros, 2016*).

## Statistics

The Shapiro-Wilk test for normality showed that CNV lesions show no normal distribution. To account for this and to consider comparisons between multiple groups, CNV lesion analyses between multiple experimental groups were performed with a Kruskal-Wallis test followed by Dunn's test. When only two groups were present ($Vegfa^{hyper}C3^{+/+}$ mice *versus* $Vegfa^{hyper}C3^{-/-}$ mice) a Mann-Whitney test (two-sided) was performed. P-values<0.05 were considered to be statistically significant. Graphs and analyses were performed with GraphPad Prism version 8.4.3.

## Acknowledgements

This work was supported by grants from the NIH (R21EY027104) and the BrightFocus foundation to AGM. JM was supported by a scholarship of the Hans-Boeckler-Stiftung. AA was supported by an Alpha Omega Alpha Carolyn L Kucklein Student Research Fellowship. We would like to thank Dr. Mark Vangel (Massachusetts General Hospital) for advice on biostatistical analyses (supported by Harvard Catalyst).

## Additional information

### Funding

| Funder | Grant reference number | Author |
| --- | --- | --- |
| National Institutes of Health | R21EY027104 | Joseph O Lamontagne<br>Karin Strittmatter<br>Alexander G Marneros |
| BrightFocus Foundation | | Alexander G Marneros |
| Hans-Böckler-Stiftung | Scholarship | Jakob Malsy |
| Alpha Omega Alpha | Carolyn L. Kucklein Student Research Fellowship | Andrea C Alvarado |

The funders had no role in study design, data collection and interpretation, or the decision to submit the work for publication.

### Author contributions

Jakob Malsy, Data curation, Formal analysis, Investigation; Andrea C Alvarado, Karin Strittmatter, Investigation; Joseph O Lamontagne, Investigation, participated in experiments as part of this revision; Alexander G Marneros, Conceptualization, Resources, Data curation, Formal analysis, Supervision, Funding acquisition, Validation, Investigation, Visualization, Methodology, Writing - original draft, Project administration, Writing - review and editing

### Author ORCIDs

Alexander G Marneros (iD) https://orcid.org/0000-0003-3866-020X

### Ethics

Animal experimentation: For all animal studies, institutional approval by Massachusetts General Hospital was granted (IACUC approval 2009N000083). Experiments were performed in compliance with ARRIVAL guidelines.

### Decision letter and Author response

Decision letter https://doi.org/10.7554/eLife.60194.sa1
Author response https://doi.org/10.7554/eLife.60194.sa2

## Additional files

### Supplementary files

• Transparent reporting form

### Data availability

All data generated or analysed during this study are included in the manuscript and supporting files.

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
