## [Decision Letter]

**Acceptance summary:**

The strength of this study is the application of various conditional and global genetically modified mice. With these tools, the authors were able to dissect the role of complement- and inflammasome-mediated inflammation in choroidal neovascularization disease.

**Decision letter after peer review:**

[Editors’ note: the authors submitted for reconsideration following the decision after peer review. What follows is the decision letter after the first round of review.]

Thank you for submitting your work entitled "Distinct effects of complement and of NLRP3- and non-NLRP3 inflammasomes for choroidal neovascularization" for consideration by *eLife*. Your article has been reviewed by three peer reviewers, and the evaluation has been overseen by a Reviewing Editor and a Senior Editor. The reviewers have opted to remain anonymous.

Our decision has been reached after consultation between the reviewers. Based on these discussions and the individual reviews below, we regret to inform you that your work will not be considered further for publication in *eLife*. The reviewers all indicated that the current manuscript is of potential to the field. However, various key concerns regarding interpretation were raised based on needed validation steps of some of the reagents and approaches, such as the new mouse lines generated, markers used for endothelial cells versus RPE, and uncertainty of comparison with caspase expression. Missing points of discussion were also noted, for example age-related effects of C3 knockout and inflammasome-independent role for complement in AMD. A disconnect between the complement aspects of the manuscript and the inflammasome was raised as well.

Reviewer #2:

This study examined the relationship and contribution of inflammasomes and complement in CNV through a spontaneous *VEGF-A^hyper^* induced AMD/CNV model, in combination with various mice models deficient in caspase 1, caspase 11, *nlrp3* or c3 using immunofluorescence staining, western blot and IHC techniques. The authors show that *Nlrp3* inflammasome activation mainly in macrophages, but not in RPE cells, contributes to CNV progression; not only *Nlrp3*-dependent, but also *Nlrp3*-independent inflammasomes promote CNV; non-canonical inflammasome activation does not induce CNV; complement and inflammasomes induce CNV independently. The findings could be very interesting and important for understanding of AMD pathology and suggest strategies of therapy. However, some of the conclusions are not fully supported by data.

1) Data shown in Figure 2B (male and female combined data of CNV area) is contradictory to 2D. In B, there is no difference between *VEGF-A^hyper^* mice and *VEGF-A^hyper^VMD2^Cre+/^WTNLRP3^A350V/A350V^* (which lack *Nlrp3* in all non-RPE cells, P=0.9483). In D, both male and female show a 40-50% reduction of CNV area in *VEGF-A^hyper^VMD2^Cre+/^WTNLRP3^A350V/A350V^* compared to *VEGF-A^hyper^* mice (female, P=0.0287; male, P=0.0195). Please clarify.

2) From Figure 4C, the authors intended to conclude that "non-canonical inflammasome activation (mediated by casp11 instead of casp1) does not affect CNV lesions in *VEGF-A^hyper^* mice". They showed that *VEGF-A^hyper^* mice lacking both casp1 and casp11 had reduced number and size of CNV, and mice lacking only casp11 did not. Nonetheless, there was no data using mice lacking casp1 only to draw the conclusion. Confusingly, the authors did mention: "Notably, lack of caspase-1 potently reduces CNV lesion numbers in both male and female mice (Figure 4C)", but this (*VEGF-A^hyper^casp1^-/-^*) cannot be found.

3) It is shown that C3^-/-^ did not cause damage in the retina (IHC) but reduced the CNV numbers in *VEGF-A^hyper^* mice of 6-weeks old (Figure 5D and B), while western blot (Figure 5C) shows C3^-/-^ induces inflammasome activation (p10 upregulated by 3.85 times compared to WT, 15-17 months old). This suggests that although C3 inhibition is safe in young mice, but it may be deleterious in old mice. This need be discussed.

4) Western blot in Figure 4A and B: Is one RPE/cho lysate sample from each strain at indicated age shown in the blot? Given the likely variation of protein expression among eyes/animals, N number should be given in figure legends or Materials and methods.

Reviewer #3:

Here, the authors use several unique mouse models to better describe how the inflammasome and complement affect the progression of AMD-like pathology in a mouse model of disease. In regard to the inflammasome, the authors clear up a known point of contention by revealing in their model that the inflammasome within RPE does not necessarily contribute to disease. The authors show that C3, from the complement cascade, does play a role an inflammasome independent role in disease; however, this was only seen in male mice. Overall, the manuscript has strong power in the numbers of samples tested, but there is a lack of depth in the results, which will be described below.

1) The authors make a convincing claim that non-RPE cells are responsible for the inflammation associated with inflammasome activation. However, they conclude macrophages are the primary source of inflammasome-mediated inflammation. This conclusion is supported through the colocalization of active caspase 1 and F4/80. It is well-known that macrophages and retina-resident microglia express F4/80 at high and low levels, respectively. Thus it is very difficult or impossible to distinguish between the two using only F4/80 and confocal microscopy. The authors should differentiate between the two populations.

2) There is a disconnect between the inflammasome part of the manuscript and the complement part of the manuscript. More directly, the complement part seems like it does not belong to this manuscript. A more appropriate figure would be to identify the factor that is triggering inflammasome activation. I recognize that a hypothesis may have been that C3 is the trigger; however, that is clearly not the case.

Reviewer #4:

In this study, the authors addressed four questions related to the pathogenesis of CNV: (1) Can NLRP3 inflammasome activation promote CNV? (2) Does NLRP3 inflammasome activation in RPE or non-RPE promote CNV? (3) Does non-canonical inflammasome activation promote CNV? (4) Does complement pathway activation promote CNV via activation of inflammasomes? These are all important questions and the authors used various genetic modified mice to address these questions.

I have a few concerns.

The model: The authors used the *VEGF-A^Hyper^* mice as a model of CNV. The authors have published a few papers in this model – all nicely demonstrated spontaneous development of CNV in these mice. In this study, the authors stained RPE/choroidal flatmounts with phalloidin and anti-CD31. Phalloidin labelling differentiates normal RPE from damaged/degenerated RPE, whereas CNV (neovascularization) is identified by anti-CD31 staining. RPE damage and CNV are two different types of lesions. Although CNV will certainly damage RPE, the damages can also occur in the absence of CNV. Various models of GA are examples of RPE degeneration without CNV (e.g., AluRNA mediated inflammasome activation induced mouse model of GA). I notice some lesions in Figures 4D and 5B appear to be absent of CNV. I suspect that the *VEGF-A^Hyper^* mice may develop both GA and CNV or perhaps the CNV is secondary to GA. The authors used phalloidin labelling to identify/measure "CNV" lesion (Figure 1A), but the lesions are in fact "RPE damages". The In my view, before the authors can accurately interpret their results it is important for them to fully understand the model.

The phenotype of genetically modified mice: The authors cross their *VEGF-A^Hyper^* mice with various other GM mice to generate TG/knockout (KO) or TG/double knockout (DKO) mice. However, the authors failed to present any data to prove the deletion of the targeted gene in the newly developed TG/KO or TG/DKO mice. For example, the authors failed to present any evidence to show that the *VEGF-A^hyper^VMD2^Cre+^/WTNLRP3^A350V/A350V^*mice had constitutive NLRP3 activation in RPE cells but not in other cells. They also did not show that RPE cells in the *VEGF-A^hyper^VMD2^Cre-^NLRP3^A350V/A350V^* mice did not express any NLRP3. The phenotype of the *VEGF-A^Hyper^caspase-1^-/-^caspase-11^-/-^, VEGF-A^Hyper^caspase-11^-/-^*, and *VEGF-A^Hyper^C3^-/-^* mice should also be checked.

Other comments:

The authors presented an age-dependent caspase-1 activation in RPE/choroid in *VEGF-A^Hyper^* mice (Figure 4A). Did increased caspase-1 activation lead to worsening of RPE/choroidal lesion during aging in the *VEGF-A^Hyper^* mice?

CVN lesions were examined in 6-week old mice, whereas NLRP3 and caspase-1 activation were examined in mice aged 2 months and older. In order to support their conclusions that NLRP3 inflammasome activation in non-RPE cells and NLRP-3-dependent and caspase-11-independent inflammasome activation pathways are involved in CNV lesion formation in the *VEGF-A^Hyper^* mice, the authors must either investigate these pathways (in WB) at 6 weeks or examine the lesions at 2 months.

Evidence of complement activation at the lesion site in the *VEGF-A^Hyper^* mice and lack of complement activation in the *VEGF-A^Hyper^ C3^-/-^*mice should be presented. Also, complement activation involves a number of enzymatic cascade. The authors should specify in their hypothesis which part of complement protein (fragment) may induce inflammasome activation in CNV, and this complement protein(fragment) should be examined in the *VEGF-A^Hyper^* and *VEGF-A^Hyper^ C3^-/-^* mice.

The authors addressed the four questions using solely GM mice. They have not investigated any of the pathways using other methods e.g. pharmacological blockade or in vitro models. As in GM mice, deletion of one gene may lead to alterations in various other genes to compensate the deleted gene, the lack of significant change in lesion development may not necessarily suggest the no-involvement of the gene to the disease. For this reason, verification of the results using other methods is necessary, particularly for these important questions.

[Editors’ note: further revisions were suggested prior to acceptance, as described below.]

Thank you for submitting your article "Distinct effects of complement and of NLRP3- and non-NLRP3 inflammasomes for choroidal neovascularization" for consideration by *eLife*. Your article has been reviewed by two peer reviewers, and the evaluation has been overseen by a Reviewing Editor and Carla Rothlin as the Senior Editor. The reviewers have opted to remain anonymous.

The reviewers have discussed the reviews with one another and the Reviewing Editor has drafted this decision to help you prepare a revised submission.

We would like to draw your attention to changes in our revision policy that we have made in response to COVID-19 (https://elifesciences.org/articles/57162). Specifically, when editors judge that a submitted work as a whole belongs in *eLife* but that some conclusions require a modest amount of validation data, as they do with your paper, we are asking that the manuscript address these points, be revised to either limit claims to those supported by data in hand, or to explicitly state that the relevant conclusions require additional supporting data.

General Comments:

The authors were very responsive to the previous concerns raised by the reviewers. There are still several outstanding concerns. It is requested that direct demonstration of NLRP3 activation in RPE but not non-RPE cells in the *Vegf-^hyper^*; *Best1-^Cre^*; NLRP3 mice is included. Validation of Caspase11 KO was also requested. Lastly, there are statistical concerns to address.

Reviewer #3:

Here, the authors have made their manuscript stronger with additional confocal images to better characterize the F4/80+ cells that contribute to caspase 1 activity. They have increased the quality of the discussion to place their results into context among the previous literature that investigates retinal inflammation and pathology. For these reasons, the manuscript is written and presented in a manner that is suitable for *eLife*.

Reviewer #4:

As the authors highlighted in their rebuttal, one of the strengths of this study is to use various GM mice to dissect the role of specific inflammatory pathways. It is therefore, important to ensure the phenotype of each GM mouse lines used in the study. In response to my comment on the phenotypes of their in-house generated TG/KO/DKO mice, the authors provided additional data to support the lack of complement activation in the *VEGF-A^Hyper^ C3^-/-^* mice. This additional data confirmed the phenotype of the mouse line.

The authors failed to prove that NLRP3 is constitutively activated in RPE cells of the *Vegfa^hyper^Best1^Cre^/+Nlrp^A350V/A350V^* mice.

The authors claimed that "we established mice that have constitutive activation of NLRP3 only in the RPE, whereas NLRP3 is inactivated in all other cells of these mice (Figure 2A)". Figure 2A only showed Cre expression in RPE flatmount. The authors used *Vegfa^hyper^Best1^Cre^/+Nlrp^A350V/A350V^* mice and the *Vegfa^hyper^Nlrp^-/-^* mice to address the question of whether NLRP3 inflammasome activation in RPE or non-RPE promote CNV. The manuscript contains no evidence of "NLRP3 activation only in RPE cells but not in all other cells in the *Vegfa^hyper^Best1^Cre^/+Nlrp^A350V/A350V^* mice".

Capase-11 expression should also be checked in their *VEGF-A^Hyper^caspase-1^-/-^caspase-11^-/-^* and *VEGF-A^Hyper^caspase-11^-/-^* mice.

---

## [Author Response]

[Editors’ note: The authors appealed the original decision. What follows is the authors’ response to the first round of review.]

Reviewer #2:This study examined the relationship and contribution of inflammasomes and complement in CNV through a spontaneous VEGF-A^hyper^ induced AMD/CNV model, in combination with various mice models deficient in caspase 1, caspase 11, nlrp3 or c3 using immunofluorescence staining, western blot and IHC techniques. The authors show that Nlrp3 inflammasome activation mainly in macrophages, but not in RPE cells, contributes to CNV progression; not only Nlrp3-dependent, but also Nlrp3-independent inflammasomes promote CNV; non-canonical inflammasome activation does not induce CNV; complement and inflammasomes induce CNV independently. The findings could be very interesting and important for understanding of AMD pathology and suggest strategies of therapy. However, some of the conclusions are not fully supported by data.1) Data shown in Figure 2B (male and female combined data of CNV area) is contradictory to 2D. In B, there is no difference between VEGF-A^hyper^ mice and VEGF-A^hyper^VMD2^Cre+^/WTNLRP3^A350V/A350V^ (which lack Nlrp3 in all non-RPE cells, P=0.9483). In D, both male and female show a 40-50% reduction of CNV area in VEGF-A^hyper^VMD2^Cre+^/WTNLRP3^A350V/A350V^ compared to VEGF-A^hyper^ mice (female, P=0.0287; male, P=0.0195). Please clarify.

For CNV lesions numbers and CNV lesion areas there is no significant difference between *Vegfa^hyper^Nlrp3^-/-^* mice (lacking NLRP3 in all cells) compared to *Vegfa^hyper^Best1^Cre/+^Nlrp3^A350V/A350V^* mice (lacking NLRP3 in all cells but with constitutive activation of NLRP3 in RPE cells only). Thus, NLRP3 activity in RPE cells does not have a major influence on CNV lesion formation. Both strains inhibit CNV lesion numbers compared to *Vegfa^hyper^*mice. The reviewer is correct that Figure 2C contained an oversight that we have now corrected.

2) From Figure 4C, the authors intended to conclude that "non-canonical inflammasome activation (mediated by casp11 instead of casp1) does not affect CNV lesions in VEGF-A^hyper^ mice". They showed that VEGF-A^hyper^ mice lacking both casp1 and casp11 had reduced number and size of CNV, and mice lacking only casp11 did not. Nonetheless, there was no data using mice lacking casp1 only to draw the conclusion. Confusingly, the authors did mention: "Notably, lack of caspase-1 potently reduces CNV lesion numbers in both male and female mice (Figure 4C)", but this (VEGF-A^hyper^casp1^-/-^) cannot be found.

We explain in the manuscript that caspase-1 KO mice with normal caspase-11 are not available as they have an incidental caspase-11 mutation as well (Casp1 and Casp11 are too close in the genome to be segregated by recombination; consequently, the published *Casp1^-/-^* mice lack both caspase-11 and caspase-1 [Kayagaki et al., 2007]).

Thus, caspase-1 KO mice always also have a caspase-11 inactivation. In order to assess whether the reduction in CNV lesions in these *Casp1^-/-^/Casp11^-/-^* double KO mice is in part explained by lack of caspase-11, we included *Casp11^-/-^* mice that have WT levels of caspase-1 as comparisons. As isolated caspase-11 deficiency in the caspase-11 KO mice does not affect CNV lesion formation, we conclude that the reduction in CNV is due to caspase-1 deficiency and not caspase-11 deficiency. We corrected the statement that the reviewer refers to and describe these mice throughout the manuscript as *Casp1^-/^/Casp11^-/-^* double KO mice.

We write: “Mice that lack caspase-1 (thus, lacking all canonical inflammasomes as caspase-1 is a required component of all inflammasomes) have also an incidental inactivating mutation in caspase-11 (they are therefore double KOs for both caspase-1 and caspase 11 [referred to as *Vegfa^hyper^Casp1^-/-^Casp11^-/-^* mice; Kayagaki et al., 2011]. We show that these *Vegfa^hyper^Casp1^-/^Casp11^-/-^* mice have indeed no caspase-1 protein [thus, no inflammasome activity], as no pro-caspase-1 nor any caspase-1 p10 can be detected in western blots of RPE/choroid lysates from these mice [Figure 4B]). To rule out any contribution of caspase-11 to inflammasome activation or CNV lesion formation in *Vegfa^hyper^* mice, we generated *Vegfa^hyper^Casp11^-/-^* mice, which have a functional caspase-1 gene (thus, can have canonical inflammasome activation) but lack caspase-11.”

3) It is shown that C3+ did not cause damage in the retina (IHC) but reduced the CNV numbers in VEGF-A^hyper^ mice of 6-weeks old (Figure 5D and B), while western blot (Figure 5C) shows C3^-/-^ induces inflammasome activation (p10 upregulated by 3.85 times compared to WT, 15-17 months old). This suggests that although C3 inhibition is safe in young mice, but it may be deleterious in old mice. This need be discussed.

We include now new choroidal flat mount images from aged *C3^-/-^* mice (12-months old), showing that these mice do not have a disruption of RPE cell morphology and do not form CNV lesions (Figure 5—figure supplement 2). Notably, whereas C3 deficiency does not result in abnormal RPE and retina at 6-weeks of age, C3 deficiency affects retinal function in aged mice. Detailed previously published studies have described these abnormalities, suggesting that C3 has a role for retinal homeostasis during aging. We have included now a discussion of these findings and of the previous eye studies on aged *C3^-/-^* mice.

4) Western blot in Figure 4A and B: Is one RPE/cho lysate sample from each strain at indicated age shown in the blot? Given the likely variation of protein expression among eyes/animals, N number should be given in figure legends or Materials and methods.

This information was provided in the figure legends. Figure 4A: “Four RPE/choroid tissues from four independent mice were pooled for each of the groups indicated.” Figure 4B: “Lysates of RPE/choroid tissues from 12-14 months-old mice (four RPE/choroids pooled for each group).”

Reviewer #3:Here, the authors use several unique mouse models to better describe how the inflammasome and complement affect the progression of AMD-like pathology in a mouse model of disease. In regard to the inflammasome, the authors clear up a known point of contention by revealing in their model that the inflammasome within RPE does not necessarily contribute to disease. The authors show that C3, from the complement cascade, does play a role an inflammasome independent role in disease; however, this was only seen in male mice. Overall, the manuscript has strong power in the numbers of samples tested, but there is a lack of depth in the results, which will be described below.1) The authors make a convincing claim that non-RPE cells are responsible for the inflammation associated with inflammasome activation. However, they conclude macrophages are the primary source of inflammasome-mediated inflammation. This conclusion is supported through the colocalization of active caspase 1 and F4/80. It is well-known that macrophages and retina-resident microglia express F4/80 at high and low levels, respectively. Thus it is very difficult or impossible to distinguish between the two using only F4/80 and confocal microscopy. The authors should differentiate between the two populations.

Our immunolabelings show clear co-localization of the inflammasome activation marker cleaved caspase-1 p10 with F4/80. This is why we stated throughout the manuscript that inflammasome activation occurs in F4/80^+^ “non-RPE cells”. We did not mean to conclude that monocyte-derived macrophages are the only cell type in CNV lesions in which inflammasome activation occurs, but rather that inflammasome activation occurs mainly in F4/80^+^ cells, which could be both monocyte-derived macrophages as well as retinal microglia.

To address the question whether these F4/80^+^ cells that infiltrate CNV lesions in *Vegfa^hyper^* mice and that show strong immunolabeling for caspase-1 p10 are monocyte-derived macrophages or activated retinal microglia cells, we performed now co-immunolabeling experiments with antibodies against the microglia marker P2RY12 and antibodies against F4/80 and caspase-1 p10. We find (see new Figure 3C) that in CNV lesions of *Vegfa^hyper^* mice a subset of F4/80^+^p10^+^ cells express P2RY12, whereas other F4/80^+^p10^+^ cells do not show immunolabeling for P2RY12. Thus, these findings suggest that both monocyte-derived macrophages (F4/80^+^p10^+^P2RY12^neg^) as well as activated retinal microglia cells (F4/80^+^p10^+^P2RY12^+^) that infiltrate the site of CNV lesion formation show inflammasome activation (caspase-1 p10^+^). Notably, retinal microglia cells at sites of the retina devoid of CNV lesions show no p10^+^ immunolabeling (Figure 3—figure supplement 1). These findings demonstrate that inflammasome activation occurs in activated retinal microglia cells once they infiltrate CNV lesions but not in nonlesional microglia cells that are not activated. These novel data are a critical new finding that we discuss in the paper. We also clarify throughout the manuscript that inflammasome activation is detected primarily in F4/80^+^ non-RPE cells that likely constitute both monocyte-derived macrophages and retinal microglia cells and not just macrophages.

2) There is a disconnect between the inflammasome part of the manuscript and the complement part of the manuscript. More directly, the complement part seems like it does not belong to this manuscript. A more appropriate figure would be to identify the factor that is triggering inflammasome activation. I recognize that a hypothesis may have been that C3 is the trigger; however, that is clearly not the case.

Several publications based only on *in vitro* data have proposed that complement pathway activation can induce inflammasome activation. As genetic studies in patients with AMD have provided a clear link between complement pathway activation and AMD pathogenesis, this has raised the important question in the field whether complement activation products may be a major inducer of inflammasome activation in CNV. In *Vegfa^hyper^* mice we also found the complement activation product C5b-9 in CNV lesions (Marneros et al., 2016), further raising the intriguing question whether inflammasome activation in these CNV lesions is at least in part due to complement activation. We have previously already identified other known triggers of the NLRP3 inflammasome that are also present in human AMD lesions to be highly enriched in CNV lesions of these mice, including increased oxidative stress and sub-RPE lipid deposits (Marneros et al., 2013).

Answering the important question whether complement activation products are major inducers of inflammasome activation also *in vivo* in CNV lesions and not only in artificial in vitro systems that do not reflect the complex interplay of many cell types and factors that are present in CNV lesions, has significant clinical relevance regarding the development of treatment strategies that inhibit both inflammatory pathways. Our data provide now *in vivo* evidence that inflammasome activation and complement pathway activation promote CNV largely independently of each other. This is an important finding and puts previous *in vitro* findings into perspective. Thus, we feel that the complement data in the paper are not disjointed from the other findings in the manuscript.

We have now added new data (e.g. we show that C3 also accumulates in CNV lesions of *Vegfa^hyper^* mice) and improved the part of the manuscript that describes the role of complement C3 in CNV lesion formation in *Vegfa^hyper^* mice. We have also made changes

to better connect the inflammasome part with the complement part of the manuscript and explain the relevance of our findings in greater detail. Furthermore, we added a paragraph explaining that we think that multiple factors that accumulate at sites of evolving CNV lesions in these mice promote inflammasome activation, including increased oxidative damage and sub-RPE lipid deposits.

Reviewer #4:I have a few concerns.The model: The authors used the VEGF-A^Hyper^ mice as a model of CNV. The authors have published a few papers in this model – all nicely demonstrated spontaneous development of CNV in these mice. In this study, the authors stained RPE/choroidal flatmounts with phalloidin and anti-CD31. Phalloidin labelling differentiates normal RPE from damaged/degenerated RPE, whereas CNV (neovascularization) is identified by anti-CD31 staining. RPE damage and CNV are two different types of lesions. Although CNV will certainly damage RPE, the damages can also occur in the absence of CNV. Various models of GA are examples of RPE degeneration without CNV (e.g., AluRNA mediated inflammasome activation induced mouse model of GA). I notice some lesions in Figures 4D and 5B appear to be absent of CNV. I suspect that the VEGF-A^Hyper^ mice may develop both GA and CNV or perhaps the CNV is secondary to GA. The authors used phalloidin labelling to identify/measure "CNV" lesion (Figure 1A), but the lesions are in fact "RPE damages". The In my view, before the authors can accurately interpret their results it is important for them to fully understand the model.

We have previously performed a very detailed analysis of the age-dependent AMD pathologies of these *Vegfa^hyper^* mice that we have studied for over 10 years (Marneros et al., 2013; Ablonczy et al., 2014; Marneros et al, 2016). In doing so, we have looked at hundreds of eyes and thousands of CNV lesions in mice of different ages and we find that in 6weeks-old *Vegfa^hyper^* mice the disruption of the honeycomb pattern RPE morphology (shown by phalloidin staining of choroidal flat mounts) occurs only at sites of CNV lesion formation with CD31^+^ neovessels. We previously reported that at 6-weeks of age these mice do not show GA-like RPE damage, but only at an advanced age, which is also a reason why we assessed CNV lesions at 6 weeks of age (Marneros et al., 2013). Our published findings and our additional new data that we provide as part of this revision allow us to make accurate CNV quantifications and reach solid conclusions about CNV lesion formation in this mouse model of AMD. Thus, we can provide clear data to answer this reviewer’s comment beyond any doubt.

As shown in Figure 1, for all analyses we always co-labeled choroidal flat mounts for CD31 (endothelial cell marker) and phalloidin (outlining particularly RPE membranes). When quantifying CNV lesions, each lesion was assessed at high magnification and all CNV lesions had a clear co-localization of CD31^+^ protruding neovessels with disruption of the typical honeycomb pattern of RPE cells (increased phalloidin signal). Thus, our quantifications do not reflect sites of RPE damage without CNV, but actual CD31^+^ neovascular CNV lesions. All CNV lesions in the figures mentioned by the reviewer actually also show CD31^+^ CNV lesions. It may not be apparent due to the low magnification of these images and the stronger intensity of the green color for phalloidin (we showed these images at lower magnification to demonstrate the effects on overall CNV lesion numbers in these eyes).

We have now made several additions in the revised manuscript to clearly demonstrate that RPE disruption occurs only at sites of CD31^+^ neovessel protrusion into the sub-RPE space.

These include several new images in Figure 1. We show that the typical honeycomb pattern of the RPE in choroidal flat mounts is only disrupted at sites of CD31^+^ neovessel formation, in immunolabeling experiments with antibodies against CD31 and active b-catenin (outlines RPE cell membranes) or phalloidin (Figures 1A and 1B). Sections show sub-RPE location of CD31 neovessels that originate from the choroidal vessels and are covered by RPE (Figure 1C).

The CNV flat mount image from a 6-weeks-old *Vegfa^hyper^* mouse in Figure 1 in which we outlined CNV lesions is now shown also for the red (CD31) and green (phalloidin) channels separately, making it easier to see that only at sites of CD31^+^ neovessel formation RPE cells show a loss of their regular morphology, whereas adjacent non-lesional RPE cells show a regular morphology (Figure 1D). Even in aged *Vegfa^hyper^* mice (at 12-months of age, when RPE/photoreceptor atrophy occurs in these mice [Marneros et al., 2013]) we find that the typical RPE morphology is maintained in choroidal flat mounts at sites devoid of CNV and that phalloidin accurately outlines the area of CD31^+^ neovessel protrusion (Figure 1E). This image also shows that with progressive age CNV lesions enlarge and become confluent (similarly as we have previously published [Marneros et al., 2013]). Finally, we also show the images that the reviewer mentioned (Figure 4D) not only as merged (red/green) images but also in their respective individual channels (red CD31 staining; green phalloidin), thereby, clearly demonstrating that RPE disruption occurs only at sites of CNV (Figure 4—figure supplement 1). Thus, our quantifications accurately measure CNV lesions at areas where CD31^+^ neovessels protrude into the sub-RPE space. We have made changes in the Results part to make this clear as well.

The phenotype of genetically modified mice: The authors cross their VEGF-A^Hyper^ mice with various other GM mice to generate TG/knockout (KO) or TG/double knockout (DKO) mice. However, the authors failed to present any data to prove the deletion of the targeted gene in the newly developed TG/KO or TG/DKO mice. For example, the authors failed to present any evidence to show that the VEGF-A^hyper^VMD2^Cre+/^WTNLRP3^A350V/A350V^ mice had constitutive NLRP3 activation in RPE cells but not in other cells. They also did not show that RPE cells in the VEGF-A^hyper^VMD2^Cre-^NLRP3^A350V/A350V^ mice did not express any NLRP3. The phenotype of the VEGF-A^Hyper^caspase-1^-/-^caspase-11+, VEGF-A^Hyper^caspase-11^-/-^, and VEGF-A^Hyper^C3^-/-^ mice should also be checked.

We show in western blots with RPE/choroid lysates of *Vegfa^hyper^Nlrp3^-/-^* mice ( = *Vegfa^hyper^ Best1^Cre/+^Nlrp3^A350V/A350V^* mice) that they indeed completely lack NLRP3 protein by using a validated anti-NLRP3 antibody, whereas NLRP3 protein is detected in RPE/choroid lysates from *VEGF-A^hyper^* mice and WT mice (Figures 4A and 4B). We also show in these western blots using anti-caspase-1 antibodies that RPE/choroid lysates of *Vegfa^hyper^Casp1^-/-^Casp11^-/-^* mice do not show any caspase-1 protein (neither pro-caspase-1 nor its activation product cleaved caspase-1 p10), whereas caspase-1 and p10 are strongly increased in RPE/choroid lysates of *VEGF-A^hyper^* mice compared to WT mice. That *C3^-/-^* and *Casp11^-/-^* mice do not produce C3 and caspase-11 protein respectively has already been shown in many publications that we referenced (and the genotype for each mouse used in this study was confirmed). Furthermore, we add now new immunolabeling data (in Figures 5A and 5B) showing that complement C3 accumulates along Bruch’s membrane and in CNV lesions of *Vegfa^hyper^* mice, whereas no C3 is observed in eyes and CNV lesions of *Vegfa^hyper^C3^-/-^* mice (confirming that *C3^-/-^* mice do not produce any complement C3).

We have previously published a detailed analysis of *Best1^Cre/+^* ( = *VMD2^Cre/+^* mice) and confirmed that this strains expresses Cre recombinase only in RPE cells in the eye (He et al., 2014), which we also confirmed in this study (Figure 2A; Figure 2—figure supplement 1).

Cre-mediated excision of the floxed neo*R* cassette in *neoR^fl/fl^-Nlrp3^A350V/A350V^* mice leads to constitutively active NLRP3, which has been shown in various different Cre strains (Brydges et al., 2009). We stained choroidal flat mounts of *Vegfa^hyper^Best1^Cre/+^Nlrp3^A350V/A350V^* mice (lacking NLRP3 in all cells but with constitutive activation of NLRP3 in RPE cells only due to the RPE-specific activity of the *Best1^Cre/+^* strain) for Cre and used only mice that showed uniform Cre expression in the RPE (Figure 2A). Thus, we quantified CNV lesions only in mice that showed strong and uniform RPE-specific Cre expression that leads to constitutive activation of NLRP3.

The absence of cleaved caspase-1 p10 immunolabeling in RPE cells of these mice (shown in Figure 3B) suggests that this constitutive activation of NLPR3 in RPE cells does not lead to high-level NLRP3 inflammasome activation that is detectable by immunolabeling in these cells, as constitutively active NLRP3 in the absence of caspase-1 cannot result in NLRP3 inflammasome activation. This is consistent with our conclusion that even with constitutive NLRP3 activity in the RPE, there is no significant inflammasome activation in the RPE in CNV lesions that would affect CNV lesion formation. Of note, constitutive activation of NLRP3 occurs in RPE cells of these mice not due to increased protein levels of NLRP3 but due to a single point mutation (*A350V*). Thus, western blotting of RPE cells isolated from these mice for NLRP3 would not detect a difference between normal NLRP3 expression and NLRP3^A350V^ protein. *Vegfa^hyper^Best1^Cre/+^Nlrp3^A350V/A350V^* mice show the same inhibition of CNV lesion formation as *Vegfa^hyper^Nlrp3^-/-^* mice and, therefore, provide a proof-of-principle additional line of evidence (in addition to the absence of caspase-1 immunodetection in RPE cells) that even potential low level activation of the NLRP3 inflammasome in the RPE due to constitutive NLRP3 activity that may not be detected by immunolabeling does not have a major effect on CNV lesion formation. In conclusion, our own data presented in this manuscript and previously published data confirm the validity of all the genetic mouse models used in this study.

Other comments:The authors presented an age-dependent caspase-1 activation in RPE/choroid in VEGF-A^Hyper^ mice (Figure 4A). Did increased caspase-1 activation lead to worsening of RPE/choroidal lesion during aging in the VEGF-A^Hyper^ mice?

The age-dependent increase in inflammasome activation in RPE/choroids of *Vegfa^hyper^* mice correlates with progressive exacerbation of CNV and AMD pathologies in these mice. CNV lesions become larger with progressive age and become confluent (Figure 1E) (which we also reported previously: Marneros et al., 2013; Ablonczy et al., 2014; Marneros et al., 2016). We made changes in the manuscript to emphasize this correlation.

This is why we quantified CNV lesions in young mice in order to be able to precisely count CNV lesions at an early stage before they become confluent. We stated this in the manuscript: “For all CNV quantifications across all experimental mouse groups we used 6-weeks-old mice, since at this young age early CNV lesions are present, whereas with progressive age CNV lesions increase in size and adjacent CNV lesions can fuse to form confluent larger lesions. Thus, quantifying CNV lesions in young 6-weeksold age-matched mice allows us to determine effects on early CNV lesion induction and to quantitate accurately the number of initial CNV lesions present at a defined early stage of the disease process before they become confluent larger lesions.”

We have provided in the revised manuscript additional images from aged *Vegfa^hyper^* mice (Figure 1E) showing that they have large confluent CNV lesions compared to young *Vegfa^hyper^* mice.

CVN lesions were examined in 6-week old mice, whereas NLRP3 and caspase-1 activation were examined in mice aged 2 months and older. In order to support their conclusions that NLRP3 inflammasome activation in non-RPE cells and NLRP-3-dependent and caspase-11-independent inflammasome activation pathways are involved in CNV lesion formation in the VEGF-A^Hyper^ mice, the authors must either investigate these pathways (in WB) at 6 weeks or examine the lesions at 2 months.

As requested by this reviewer, we have now included a new WB of RPE/choroid lysates from 6-weeks-old *Vegfa^hyper^* mice and their WT littermates. We show that already at that young age at which we quantified CNV lesions inflammasome activation is clearly increased in *Vegfa^hyper^* mice (>2fold increase in caspase-1 p10 protein levels) (Figure 1F).

Evidence of complement activation at the lesion site in the VEGF-A^Hyper^ mice and lack of complement activation in the VEGF-A^Hyper^ C3^-/-^mice should be presented.

We have previously shown that the complement activation product C5b-9 (for which C3 activity is required) accumulates in CNV lesions of VEGF-A^hyper^ mice (Marneros et al., 2016). To address this reviewer’s request, we show now new images (Figures 5A and 5B) that demonstrate C3 accumulation in CNV lesions of *Vegfa^hyper^* mice but complete absence of C3 in *Vegfa^hyper^C3^-/-^* mice. Thus, C3 KO mice indeed do not produce C3 protein. Moreover we show that C5b9 immunolabeling is observed in CNV lesions of *Vegfa^hyper^* mice but not in CNV lesions of *Vegfa^hyper^C3^-/-^* mice, consistent with C3 being required for C5b-9 formation (Figure 5B and Figure 5—figure supplement 1).

Also, complement activation involves a number of enzymatic cascade. The authors should specify in their hypothesis which part of complement protein (fragment) may induce inflammasome activation in CNV, and this complement protein(fragment) should be examined in the VEGF-A^Hyper^ and VEGF-A^Hyper^ C3^-/-^ mice.

*In vitro* studies have shown that the membrane attack complex C5b-9 induces sublethal damage to macrophages and thereby induces inflammasome activation. Because we observed C5b-9 accumulation in CNV lesions of *Vegfa^hyper^* mice we tested whether C3 deficiency (which results in lack of C5b-9) has an effect on inflammasome activation in CNV lesions (our hypothesis was explained in detail). We better explain in the manuscript that C5b-9 has been shown to promote inflammasome activation. We show now that both C3 and C5b-9 are present in CNV lesions of *Vegfa^hyper^* mice but absent in CNV lesions of *Vegfa^hyper^C3^-/-^* mice

The authors addressed the four questions using solely GM mice. They have not investigated any of the pathways using other methods e.g. pharmacological blockade or in vitro models. As in GM mice, deletion of one gene may lead to alterations in various other genes to compensate the deleted gene, the lack of significant change in lesion development may not necessarily suggest the no-involvement of the gene to the disease. For this reason, verification of the results using other methods is necessary, particularly for these important questions.

We have previously shown that pharmacologic blockade of inflammasomes with a specific and highly validated chemical inhibitor of caspase-1 can potently inhibit CNV *in vivo* (Marneros et al., 2016). Thus, both genetic as well as pharmacologic targeting of inflammasomes block CNV lesion formation, establishing the relevance of inflammasome activation for CNV lesion formation. Here, we use various genetic approaches to answer specific questions that cannot be answered by pharmacologic inhibitors (e.g. cell type-specific role of NLRP3 activation in RPE cells in the tissue microenvironment of evolving CNV lesions) or by *in vitro* models. Inflammasome activation in CNV lesions occurs through a complex interplay of several cells and factors that require multiple alterations in an inflammatory microenvironment and are dependent on infiltration of activated macrophages/microglia, and such a complex environment cannot be replicated *in vitro*. Thus, we do not believe that findings in artificial *in vitro* models would have real relevance regarding the questions we are addressing in this study. In fact, the strength of this study lies in the utilization of various genetic strains in a well-characterized AMD mouse model that allows us to assess how the many different cell types and factors interact in a complex tissue microenvironment to activate inflammasomes and affect CNV lesion growth. Our genetic studies show potent inhibition of CNV lesions when targeting NLRP3, caspase-1 or C3, and thus, potential upregulation of other genes are not compensating for their loss. This demonstrates unequivocally that NLRP3, caspase-1 or C3 play critical roles in VEGF-A-induced CNV lesion formation that cannot be compensated for when inactivated, making them attractive therapeutic targets.

[Editors’ note: what follows is the authors’ response to the second round of review.]

General Comments:The authors were very responsive to the previous concerns raised by the reviewers. There are still several outstanding concerns. It is requested that direct demonstration of NLRP3 activation in RPE but not non-RPE cells in the Vegf-^hyper^; Best1-^Cre^; NLRP3 mice is included. Validation of Caspase11 KO was also requested. Lastly, there are statistical concerns to address.Reviewer #4:As the authors highlighted in their rebuttal, one of the strengths of this study is to use various GM mice to dissect the role of specific inflammatory pathways. It is, therefore, important to ensure the phenotype of each GM mouse lines used in the study. In response to my comment on the phenotypes of their in-house generated TG/KO/DKO mice, the authors provided additional data to support the lack of complement activation in the VEGF-A^Hyper^ C3^-/-^ mice. This additional data confirmed the phenotype of the mouse line.The authors failed to prove that NLRP3 is constitutively activated in RPE cells of the Vegfa^hyper^Best1^Cre^/+Nlrp^A350V/A350V^ mice.The authors claimed that "we established mice that have constitutive activation of NLRP3 only in the RPE, whereas NLRP3 is inactivated in all other cells of these mice (Figure 2A)". Figure 2A only showed Cre expression in RPE flatmount. The authors used Vegfa^hyper^Best1^Cre^/+Nlrp^A350V/A350V^ mice and the Vegfa^hyper^Nlrp^-/-^ mice to address the question of whether NLRP3 inflammasome activation in RPE or non-RPE promote CNV. The manuscript contains no evidence of "NLRP3 activation only in RPE cells but not in all other cells in the Vegfa^hyper^Best1^Cre^/+Nlrp^A350V/A350V^ mice".Capase-11 expression should also be checked in their VEGF-A^Hyper^caspase-1^-/-^caspase-11^-/-^ and VEGF-A^Hyper^caspase-11^-/-^ mice.

We are now including new data showing that in RPE cells isolated from *Vegfa^hyper^Best1^Cre/+^Nlrp3^A350V/A350V^* mice Cre recombinase is not only detected in the RPE by immunolabeling but that Cre recombinase is also active in the RPE and leads to removal of the floxed *neoR* cassette in intron 2 of the *Nlrp3* gene, resulting in the mutant constitutively active *Nlrp3^A350V/A350V^* allele. Notably, no Cre-mediated excision of the floxed *neoR* cassette was observed in the retina, lens or choroid in eyes of the identical *Vegfa^hyper^Best1^Cre/+^Nlrp3^A350V/A350V^* mice and only the *Nlrp3^-/-^* allele was detected in these other ocular tissues. These findings are consistent with an RPE-specific Cre activity in eyes of *Vegfa^hyper^Best1^Cre/+^Nlrp3^A350V/A350V^*mice and that the *Nlrp3^A350V^* allele is only found in their RPE cells whereas all other eye tissues of these mice only have the *Nlrp3^-/-^* allele and are null for NLRP3. In contrast, RPE cells isolated from *Vegfa^hyper^Nlrp3^-/-^* mice have the floxed *neoR* cassette in intron 2, resulting in the inactive *Nlrp3^-/-^* allele. We added these new data to Figure 2—figure supplement 1C. We also made changes in the manuscript to better explain that the *Vegfa^hyper^Best1^Cre/+^Nlrp3^A350V/A350V^* mice have the constitutively active *Nlrp3^A350V/A350V^* allele selectively in the RPE, instead of stating that constitutive NLRP3 activation occurs in the RPE. We conclude that the constitutively active *Nlrp3^A350V/A350V^* allele in the RPE does not promote CNV lesion formation in *Vegfa^hyper^* mice. Collectively, our data provide strong evidence that CNV lesion formation in *Vegfa^hyper^* mice is not promoted by NLRP3 in the RPE but rather by NLRP3 in non-RPE cells.

That *Casp1^-/-^/Casp11^-/-^* mice and *Casp11^-/-^* mice (these are constitutive complete mutants, not conditional tissue-specific mutants) both lack functional caspase-11 protein has been shown previously (Kayagaki et al., 2011). We include this information in the revised manuscript and cite the *Nature* paper by Kayagaki et al. that showed in great detail that both these strains have no detectable functional caspase-11 protein. Genotyping confirmed that the mice in this study are the same *Casp1^-/-^/Casp11^-/-^* mice and *Casp11^-/-^* mice as in the Kayagaki et al. study.